# Copy-number and gene dependency analysis reveals partial copy loss of wild-type SF3B1 as a novel cancer vulnerability

Brenton R Paolella[1,2,3][*][†], William J Gibson[1,2,3][†], Laura M Urbanski[1,3], John A Alberta[1,4], Travis I Zack[1,2,3], Pratiti Bandopadhayay[1,2,3,5], Caitlin A Nichols[1,2,3], Pankaj K Agarwalla[6], Meredith S Brown[1,3], Rebecca Lamothe[1,3], Yong Yu[7], Peter S Choi[2,3], Esther A Obeng[2,8], Dirk Heckl[8], Guo Wei[2], Belinda Wang[2,3], Aviad Tsherniak[2], Francisca Vazquez[2], Barbara A Weir[2], David E Root[2], Glenn S Cowley[2], Sara J Buhrlage[1], Charles D Stiles[1,4], Benjamin L Ebert[2,8], William C Hahn[2,3,9], Robin Reed[7]*, Rameen Beroukhim[1,2,3,9]*

[1]Department of Cancer Biology, Dana-Farber Cancer Institute and Harvard Medical School, Boston, United States; [2]Broad Institute of Massachusetts Institute of Technology and Harvard University, Cambridge, United States; [3]Department of Medical Oncology, Dana-Farber Cancer Institute and Harvard Medical School, Boston, United States; [4]Department of Neurobiology, Harvard Medical School, Boston, United States; [5]Department of Pediatric Oncology, Dana-Farber Cancer Institute and Harvard Medical School, Boston, United States; [6]Department of Neurosurgery, Massachusetts General Hospital, Harvard Medical School, Boston, United States; [7]Department of Cell Biology, Harvard Medical School, Boston, United States; [8]Division of Hematology, Department of Medicine, Brigham and Women's Hospital, Harvard Medical School, Boston, United States; [9]Department of Medicine, Brigham and Women's Hospital and Harvard Medical School, Boston, United States

*For correspondence:
brenton_paolella@dfci.harvard.
edu (BRP); robin_reed@hms.
harvard.edu (RR);
Rameen_Beroukhim@dfci.harvard.
edu (RB)

[†]These authors contributed equally to this work

**Competing interests:** The authors declare that no competing interests exist.

**Abstract** Genomic instability is a hallmark of human cancer, and results in widespread somatic copy number alterations. We used a genome-scale shRNA viability screen in human cancer cell lines to systematically identify genes that are essential in the context of particular copy-number alterations (copy-number associated gene dependencies). The most enriched class of copy-number associated gene dependencies was CYCLOPS (Copy-number alterations Yielding Cancer Liabilities Owing to Partial losS) genes, and spliceosome components were the most prevalent. One of these, the pre-mRNA splicing factor *SF3B1*, is also frequently mutated in cancer. We validated *SF3B1* as a CYCLOPS gene and found that human cancer cells harboring partial *SF3B1* copy-loss lack a reservoir of SF3b complex that protects cells with normal *SF3B1* copy number from cell death upon partial *SF3B1* suppression. These data provide a catalog of copy-number associated gene dependencies and identify partial copy-loss of wild-type *SF3B1* as a novel, non-driver cancer gene dependency.

## Introduction

Despite recent advances in cancer therapeutics, there remains a dearth of effective treatments. Therefore, expanding the number of candidate therapeutic targets in cancer is crucial. Cancer 'driver

genes', which undergo positive selection due to their effects on oncogenes or tumor suppressor genes, represent cancer vulnerabilities that are broadly considered as potential therapeutic targets (*Cheung et al., 2011*; *Eifert and Powers, 2012*; *Wang et al., 2015*). However, alterations of non-driver genes, which do not contribute to oncogenesis but are nevertheless observed, represent an emerging class of candidate therapeutic target that have yet to be fully explored.

During the course of tumorigenesis, most cancers undergo somatic copy number alterations (SCNAs) affecting large fractions of the genome (See Appendix Note) (*Beroukhim et al., 2010*). Yet most genes affected by SCNAs likely do not contribute to oncogenesis and are therefore over-whelmingly genetically altered non-driver genes. Recently, our laboratory and others have described potential new therapeutic targets that occur as a result of SCNAs affecting non-driver genes. For example, partial copy-loss of the proteasome subunit *PSMC2*, or RNA polymerase subunit *POLR2A* sensitized cancer cells to further suppression of those genes (*Liu et al., 2015*; *Nijhawan et al., 2012*). This 'CYCLOPS' (Copy-number alterations Yielding Cancer Liabilities Owing to Partial losS) phenotype suggests that many additional cancer vulnerabilities exist as a result of SCNAs that affect non-driver genes, although some CYCLOPS genes may function as driver genes when affected by other genetic alterations besides partial copy-loss. The frequency of these CYCLOPS gene dependencies and their general features are largely unknown.

These CYCLOPS genes tend also to be cell essential genes. While essential genes would be expected to be poor therapeutic targets because of their requirement for survival in all tissues, therapeutic windows can still exist (*Muller et al., 2015*). Identifying which essential genes may be considered CYCLOPS genes, and the mechanisms underlying how normal cells tolerate partial loss of function, is necessary for developing approaches to target those therapeutic windows.

The spliceosome is one such essential protein complex that can be therapeutically targeted in cancer. Previous work suggested spliceosome components were enriched as candidate CYCLOPS genes (*Nijhawan et al., 2012*). However, spliceosome CYCLOPS dependencies have yet to be validated and the molecular mechanisms for how spliceosome CYCLOPS dependencies arise remain unclear. Compounds have been discovered that inhibit pre-mRNA splicing, with reports of broad anti-neoplastic effects (*Webb et al., 2013*). Furthermore, cancers can harbor recurrent mutations in splicing factors (*Dvinge et al., 2016*), including gain-of-function mutations in *SF3B1* (*Ellis et al., 2012*; *Harbour et al., 2013*; *Imielinski et al., 2012*; *Papaemmanuil et al., 2011*; *Wang et al., 2011*; *Yoshida et al., 2011*) that can sensitize cells to spliceosome modulatory drugs (*Obeng et al., 2016*). In addition to SF3B1 mutations, other genomic alterations in SF3B1, including copy number alterations, may also unveil novel cancer vulnerabilities. The extent to which SF3B1 and other splicing factors can be leveraged as therapeutic targets in cancer is not fully understood.

We therefore sought to systematically evaluate the prevalence of CYCLOPS dependencies relative to other SCNA-associated gene dependencies in cancer. Here, we report that CYCLOPS dependencies are the most enriched class of copy-number associated gene dependency, even more frequent than amplification of oncogene-addicted driver gene. We find that CYCLOPS genes tend to be a subset of essential genes for which there is little feedback regulation in their expression when altered by SCNAs. We also find that more CYCLOPS gene dependencies are associated with spliceosome components than with any other gene family.

We find that wild-type SF3B1 is a non-driver CYCLOPS gene dependency and describe the mechanism behind this dependency. Furthermore, the molecular mechanism of the SF3B1 CYCLOPS dependency is distinct from SF3B1 dependencies targeted by current spliceosome inhibitors. We also identify the deubiquitinase inhibitor (DUBi's), b-AP15, can reduce SF3B1 protein levels and target the SF3B1 CYCLOPS dependency. Moreover, DUBi's may represent a general therapeutic approach to target CYCLOPS gene vulnerabilities. The identification of *SF3B1* as a CYCLOPS gene highlights a previously unrecognized cancer vulnerability and implicates non-driver alterations of wild-type SF3B1 as a potential therapeutic target present in 11% of all cancers.

## Results

### Most copy-number associated cancer dependencies result from genomic loss

We interrogated copy-number associated vulnerabilities genome-wide across 179 cell lines by integrating gene dependency data from Project Achilles (*Cowley et al., 2014*) with copy-number calls for 23,124 genes (*Barretina et al., 2012*) (*Figure 1A*). The gene dependency data represented the effects on proliferation of 55,416 shRNAs targeting 11,589 unique genes, processed by the ATARiS method to estimate effects of 'on-target' shRNAs (*Shao et al., 2013*), which yielded 8724 unique gene-level dependency scores. For every pair of genes in the general analysis, we calculated Pearson correlations between the copy-number of the first gene and the dependency score of the second; yielding 201,733,776 parings in total (*Figure 1A*). We calculated p-values for each correlation and q-values to correct for multiple hypotheses (see Materials and methods). In many cases, a single gene dependency profile correlated with copy-number profiles of multiple genes from a single genomic region. We considered these to represent a single 'independently significant' interaction with the overall copy-number of that region.

In the general analysis, we identified 50 independently significant copy-number:gene-dependency interactions (q < 0.25; *Supplementary file 1A*). Approximately two-thirds (33/50) of these interactions involved genes on separate chromosomes (*trans* interactions). Among the 33 *trans* interactions, 21 reflected sensitization to suppression of a gene resulting from copy-loss of a different genomic region; the other 12 resulted from copy-gains. The *trans* gene dependencies identified were enriched for members of macromolecular complexes (q = $2.5 \times 10^{-4}$). In three cases, copy-loss of a gene (*PPP2CB*, *FUBP2*, or *MAGOH*) was associated with increased sensitivity to suppression of its paralog (*PPP2CA*, *FUBP1*, and *MAGOHB*, respectively). In contrast, all but one interaction (16/17) between genes on the same chromosome (*cis* interactions) involved increases in sensitivity to suppression of a gene that had undergone copy-loss (CYCLOPS genes) (*Nijhawan et al., 2012*). As a result, 74% of all 50 significant copy-number:gene-dependency interactions were associated with copy-loss rather than gain (p=$1.5 \times 10^{-4}$).

### Most cancers exhibit losses of candidate CYCLOPS genes

Although CYCLOPS interactions represent 0.004% (8,724/201,733,776) of all potential interactions in the general analysis, they constituted one-third of all significant interactions, making them the most enriched class of copy-number synthetic lethal interactions we identified (*Figure 1B*). Their prevalence is partly the result of frequent genomic loss in cancer genomes. Specifically, across 10,570 cancers spanning 31 cancer types profiled by The Cancer Genome Atlas (TCGA), 16% of the genome undergoes loss in the average cancer (*Figure 1—figure supplement 1A*), mainly due to losses encompassing chromosome arms or entire chromosome (*Figure 1—figure supplement 1B*). Indeed, loss of a tumor suppressor often involves such arm-level losses (*Figure 1—figure supplement 1C*). The fraction of the genome lost per tumor ranged from an average of 1.3% in thyroid cancer to 34.4% in ovarian cancer (*Figure 1C*).

To enhance the identification of CYCLOPS genes, we performed a genome-wide analysis focused on just CYCLOPS dependencies (*Figure 1D*; Materials and methods). The CYCLOPS analysis had greater power than the general analysis described above because it focused on fewer hypotheses. From the CYCLOPS analysis, we identified 124 candidate CYCLOPS genes (*Supplementary file 1B*), including 87% of the candidate CYCLOPS genes identified in the general analysis (*Supplementary file 1A*). Candidate CYCLOPS genes were distributed across all autosomes (*Figure 1E*) and were biased towards areas of frequent copy-loss (p<$10^{-15}$). The bias toward more frequent copy-loss of candidate CYCLOPS genes may reflect greater statistical power in frequently deleted regions. We also examined the reproducibility of the CYCLOPS analysis using a separate shRNA viability screen across 77 cell lines with a separate analytical approach (*Marcotte et al., 2016*) and found 49% (49/103) of the CYCLOPS genes analyzed by both datasets validated in the Marcotte dataset (p<$2 \times 10^{-4}$, binomial proportion test).

Partial copy-loss of candidate CYCLOPS genes is frequent in cancer genomes. Among the 10,570 TCGA cancers with copy-number data, 71.6% harbored loss of at least one candidate CYCLOPS

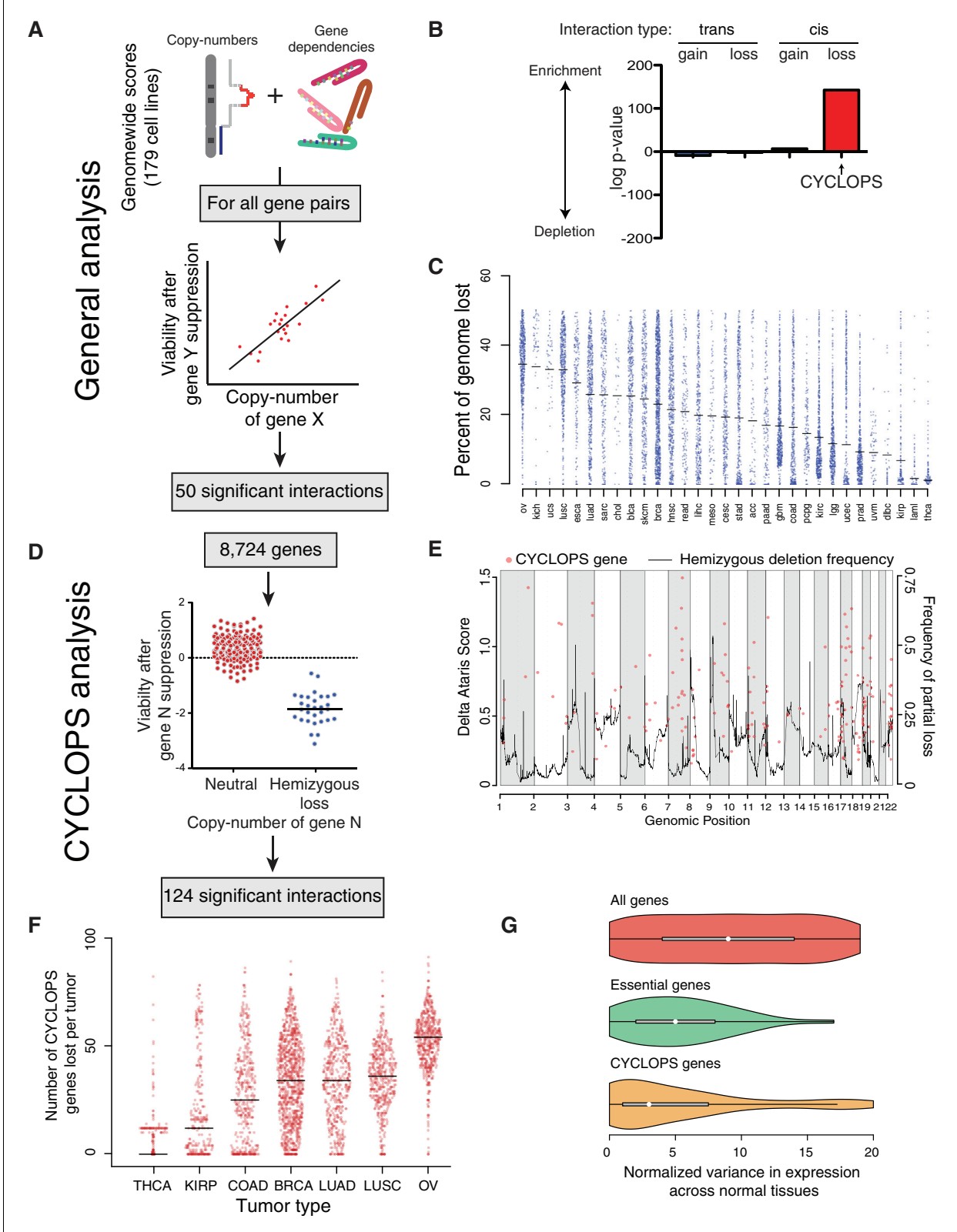

**Figure 1.** Analysis of copy-number-associated gene dependencies. (**A**) Schematic describing the general analysis of copy-number-associated gene dependencies. (**B**) Significance of enrichment for positive copy-number:gene dependency associations across *trans* and *cis* association classes, relative to the expected share assuming all possible interactions exhibited equal likelihood of positive association. (**C**) Percent of genome lost across 31 TCGA cancer types. Cancer types are indicated by TCGA abbreviations (see https://tcga-data.nci.nih.gov/datareports/codeTablesReport.htm, 'Disease Study'

eLIFE Research article

Cancer Biology | Human Biology and Medicine

*Figure 1 continued*

table). (D) Schematic describing the approach to identify CYCLOPS genes. (E) Relative strength (delta ATARiS score) of CYCLOPS genes (orange circles; left axis) and frequency of hemizygous deletion (solid black line; right axis) against genomic position (x-axis). (F) The number of CYCLOPS genes lost per tumor for various tumor types as in (C). Horizontal black lines represent medians per tumor type. (G) Distribution of variances in gene expression for different gene classes, normalized to expression level (see Materials and methods), in normal tissues. Whiskers represent min/max values and boxes represent upper and lower quartile ranges. Width of plots represents relative sample density.

The following figure supplement is available for figure 1:

**Figure supplement 1.** Analysis of copy-number-associated gene dependencies.

gene (*Figure 1F*). The average number of candidate CYCLOPS genes lost per tumor ranged from one in thyroid cancer to 47 in ovarian cancer.

## Candidate CYCLOPS genes tend to be uniformly expressed members of essential complexes

Of the 124 candidate CYCLOPS genes, 20 were members of the spliceosome and 11 were members of the proteasome, making these the only significantly enriched KEGG pathways among candidate CYCLOPS genes (*Supplementary file 1D*). Candidate CYCLOPS genes are also enriched for essential genes. Genome-wide CRISPR viability data in human cancer cell lines identified 1580 core-essential genes (*Hart et al., 2015*), including 58% (72/124) of candidate CYCLOPS genes ($p<2\times10^{-4}$, binomial proportion test).

Expression of candidate CYCLOPS genes is markedly uniform across normal tissues. Across RNA-sequencing data from 2342 samples comprising 42 tissue types (*Melé et al., 2015*), expression of candidate CYCLOPS genes varied significantly less than that of the average gene ($p=1.8\times10^{-15}$) and trended towards greater uniformity than non-CYCLOPS essential genes ($p=0.07$; *Figure 1G*).

However, expression of candidate CYCLOPS genes is highly responsive to genomic loss in cancer. We integrated expression profiles for 16,867 genes with copy-number profiles across 1011 cell lines in the Cancer Cell Line Encyclopedia (CCLE) (*Barretina et al., 2012*) to determine the influence of copy-number on gene expression. Upon copy-loss, CYCLOPS gene expression decreased by 28% on average, relative to an 18% decrease among non-CYCLOPS genes ($p<10^{-4}$, *Figure 1—figure supplement 1D–E*). These data suggest that cancers with copy-loss of CYCLOPS genes are likely to express them at lower levels than normal tissues.

## SF3B1 is a CYCLOPS gene

*SF3B1* was among the most significant candidate genes in our CYCLOPS analysis (*Supplementary file 1B*), although it was not detected in our general analysis (*Supplementary file 1A*). The SF3B1 protein is one of seven subunits (SF3B1–5, SF3B14 and PHF5A) of the SF3b complex, which is a constituent of the essential U2 snRNP splicing factor (*Wahl et al., 2009*). Cells with *SF3B1* copy-loss exhibited significantly reduced viability upon partial *SF3B1* suppression relative to cells without *SF3B1* copy-loss (mean dependency scores of −1.14 and 0.01 respectively, $p<10^{-5}$), which suggests that partial suppression of SF3B1 can be tolerated in certain contexts even though it is an essential gene.

*SF3B1* is partially lost in 11% of the 10,570 cancers from the TCGA PanCan dataset (see Materials and methods for definitions of copy number states). Across all cancers *SF3B1* copy-loss was 5.4 times more common than *SF3B1* mutations, which occur in ~2% of all cancers (*Supplementary file 1C*), and mutations and copy-loss were mutually exclusive ($p=0.007$). Losses were more frequent in breast (20%), urothelial bladder (32%) and chromophobe kidney cancers (71%). Genomic deletions of *SF3B1* typically affect most of the chromosome arm (81% of losses) and are never homozygous (0/10,570 cancers), consistent with characterization of *SF3B1* as an essential gene (*An and Henion, 2012*; *Isono et al., 2005*). In contrast, 85% of genes are homozygously deleted at least once among the same 10,570 cancers. Similarly, analysis of copy number alterations from 1042 cancer cell lines in the CCLE indicated 24% of cell lines harbor partial *SF3B1* deletion, including 16/61 (26%) of breast cancer cell lines, but never homozygous loss (0/1042 cell lines).

We established the vulnerability of SF3B1$^{loss}$ cells to SF3B1 suppression in both breast and hematopoietic lineages. We tested the proliferation of six breast lines after partial SF3B1 suppression, including three lines with SF3B1 copy-loss (SF3B1$^{loss}$) and three without either loss or gain of the gene (SF3B1$^{neutral}$). Upon partial SF3B1 suppression, SF3B1$^{loss}$ cells exhibited significant growth defects but SF3B1$^{neutral}$ cells or SF3B1$^{gain}$cells did not (**Figure 2A** and **Figure 2—figure supplement 1A**). Growth defects observed in SF3B1$^{loss}$ cells included BT549, a near triploid cell line with two copies of SF3B1. Partial SF3B1 suppression also decreased the growth of ESS1, an endometrial cell line harboring an SF3B1$^{K666N}$ mutation (**Figure 2A** and **Figure 2—figure supplement 1A**). However, complete SF3B1 suppression resulted in growth defects even in SF3B1$^{neutral}$ cells (**Figure 2—figure supplement 1B–C**), consistent with previous work establishing SF3B1 as an essential gene.

The SF3B1 shRNAs used to unveil the CYCLOPS vulnerability targeted separate regions of the gene and resulted in partial SF3B1 knockdown across all eleven breast cell lines (**Figure 2B**). However, SF3B1$^{loss}$ cells had greater levels of knockdown due to lower SF3B1 expression at baseline, which was confirmed at the protein level in four of these lines by immunoblot (**Figure 2—figure supplement 1D**).

We also generated similar results in isogenic SF3B1$^{loss}$ cells derived from the SF3B1$^{neutral}$ cell line Cal51. We generated SF3B1$^{loss}$ cells using two independent CRISPR-Cas9 mediated methods of gene editing (see Materials and methods). The first line contained a frameshift mutation inactivating one SF3B1 allele (SF3B1$^{Loss-Cal51-1}$). The second line had deletion of one copy of the SF3B1 locus, generated by co-expressing two sgRNAs: one upstream targeting a heterozygous SNP, and one downstream of SF3B1 (SF3B1$^{Loss-Cal51-2}$). In both cases CRISPR-mediated SF3B1 loss resulted in decreased proliferation upon SF3B1 suppression relative to cells that were generated in parallel but did not produce inactivating alleles (SF3B1$^{control-Cal51}$ cells; **Figure 2A** and **Figure 2—figure supplement 1A**).

We confirmed the vulnerability of the SF3B1$^{loss}$ cells to SF3B1 partial suppression using a GFP-competition assay in which we compared the proliferation rate of uninfected cells co-cultured with cells infected with a vector that co-expressed GFP and an shRNA targeting either LacZ or SF3B1. The expression of LacZ or SF3B1 shRNAs did not result in significant changes in proliferation of SF3B1$^{neutral}$ cells in seven cell lines, including the non-transformed mammary cell line, MCF10A (**Figure 2C** and **Figure 2—figure supplement 1E**). However, SF3B1$^{loss}$ cells expressing SF3B1 shRNAs were not compatible with long-term culture (**Figure 2C** and **Figure 2—figure supplement 1E**).

The SF3B1 CYCLOPS vulnerability is not recapitulated by suppression of other SF3b complex subunits, and copy number alterations of other SF3b complex genes do not confer susceptibility to SF3B1 suppression. We calculated the significance of associations between Achilles RNAi dependency data of six of the seven SF3b complex subunits (we lacked RNAi data for SF3B2) and copy numbers of the genes encoding them (**Figure 2—figure supplement 1F**). We identified associations between copy numbers of several SF3b subunits and sensitivity to suppressing that same subunit, consistent with our prior determination that multiple SF3b complex subunits are candidate CYCLOPS genes (**Supplementary file 1B**). However, we observed no associations between susceptibility to suppression of any of the SF3b complex genes and copy numbers of different SF3b subunits. Further, we confirmed one comparison where suppression of PHF5A, an SF3b complex gene, did not alter the growth of SF3B1$^{loss}$ cells (**Figure 2—figure supplement 1G–H**).

Partial suppression of SF3B1 leads to both cell cycle arrest and apoptosis in SF3B1$^{loss}$ but not SF3B1$^{neutral}$ lines. We generated cultures containing a tetracycline inducible system expressing hairpins targeting Luciferase or SF3B1 (TR-shSF3B1#3 and an additional hairpin, TR-shSF3B1#5, **Figure 2—figure supplement 1A**), enabling us to discriminate SF3B1 suppression from infection with shRNA vectors. Consistent with stable SF3B1 suppression, inducible SF3B1 suppression retards SF3B1$^{loss}$ cell growth and does not affect SF3B1$^{neutral}$ growth (**Figure 2—figure supplement 1B**) and reduces cell viability in SF3B1$^{loss}$ cells but not in SF3B1$^{neutral}$ cells (**Figure 2D and E**). SF3B1$^{loss}$ cells had significantly increased proportions of cells in G2/M phase after SF3B1 suppression, which did not occur in SF3B1$^{neutral}$ cells (**Figure 2F**). Subsequent to G2/M arrest, SF3B1$^{loss}$ cells further exhibited a significant induction in apoptosis as determined by increased number of AnnexinV/PI-positive cells that was not observed in SF3B1$^{neutral}$ cells (**Figure 2G**).

Expression of exogenous SF3B1 rescued the loss of viability in SF3B1$^{loss}$ cells, confirming the specificity of our shRNAs. We used a lentiviral construct encoding a codon-optimized SF3B1 ORF,

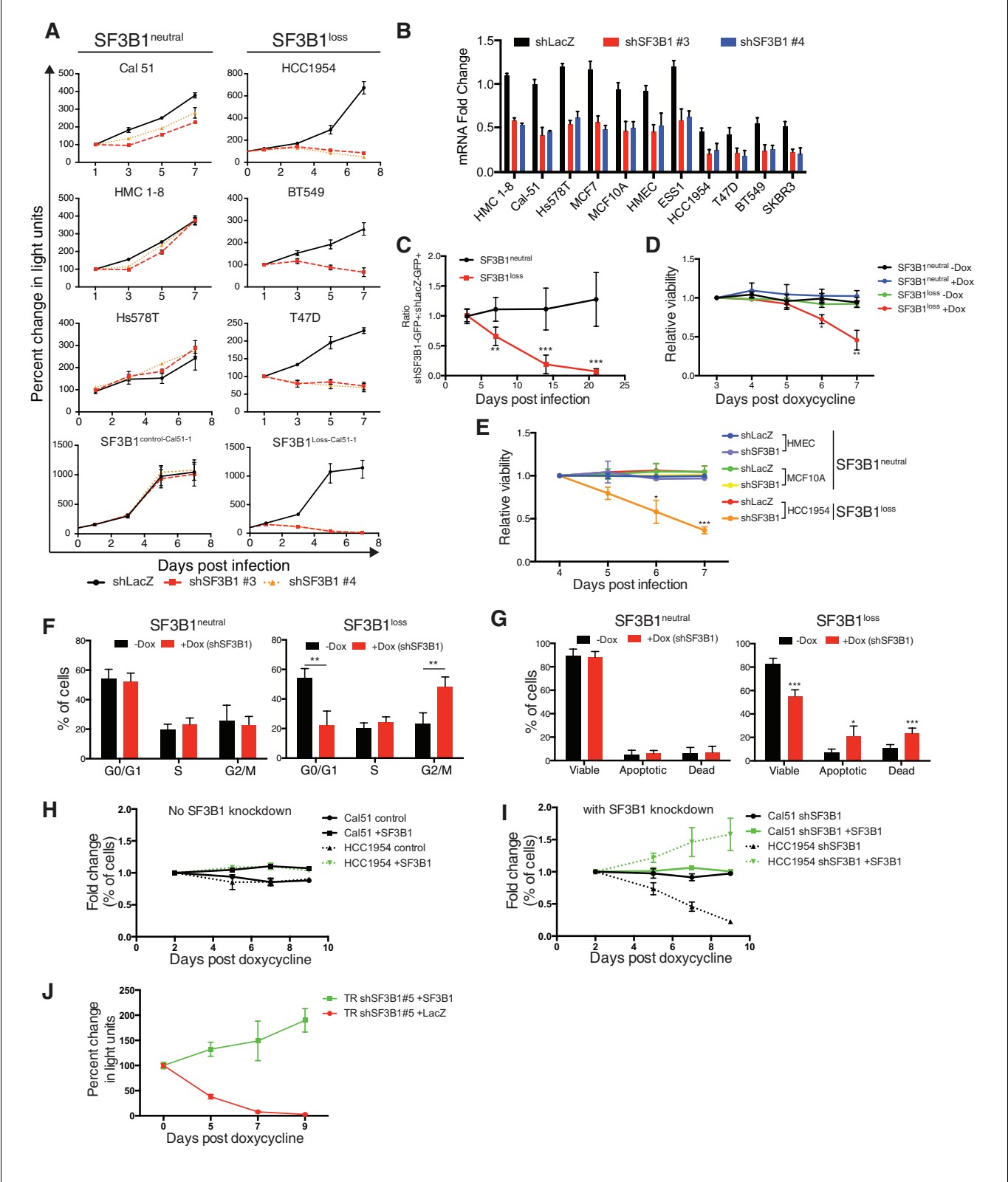

**Figure 2.** Characterization of SF3B1 as a CYCLOPS gene. (**A**) Growth of breast cancer cell lines expressing shLacZ (black) or shSF3B1 (red and orange), measured as changes in CellTiterGlo luminescence relative to one day post-infection. (**B**) Quantitative RT-PCR of *SF3B1* expression from the indicated cell lines expressing shLacZ or shSF3B1 shRNAs normalized to the diploid *SF3B1^neutral^* cell line Cal51. (**C**) Ratio of cells expressing shSF3B1-GFP relative to uninfected controls, normalized to the ratio of cells expressing shLacZ-GFP relative to uninfected controls. Data represent averages from *SF3B1^neutral^*

*Figure 2 continued on next page*

Figure 2 continued

(n = 7) and *SF3B1*<sup>loss</sup> (n = 6) cell lines, using shSF3B1 #4 in *Figure 2—figure supplement 2E*. (D) Viability of cells expressing TR-shSF3B1#3 and TRshSF3B1#5, relative to viability three days post doxycycline administration. (E) Viability of cells expressing shLacZ or the average of shSF3B1#3 and shSF3B1#4, measured asthe fraction of propidium iodide negative cells, relative to the viability of these cells four days post infection. (F) Cell cycle distribution four days after *SF3B1* suppression averaged from TR shSF3B1#3 and #5. (G) Fraction of apoptotic cells five days after *SF3B1* suppression averaged from TR shSF3B1#3 and #5, as determined by AnnexinV/PI flow cytometry. (H) Change in ratio of cells expressing SF3B1-GFP relative to uninfected cells. (I) Ratio of cells expressing SF3B1-GFP to uninfected cells, in the context of endogenous SF3B1 suppression with TR-shSF3B1 #5. (J) Growth of LacZ and SF3B1 expressing *SF3B1*<sup>loss</sup> cells upon *SF3B1* suppression (TR-shSF3B1#5), measured as changes in CellTiter-Glo luminescence. For all panels, *p<0.05 **p<0.01 ***p<0.001, and error bars represent ± SD from at least three (panels A–G) or two (H–J) replicates.

The following figure supplements are available for figure 2:

**Figure supplement 1.** Characterization of SF3B1 as a CYCLOPS gene.

**Figure supplement 2.** Further characterization of SF3B1 as a CYCLOPS gene.

which is resistant to shRNA suppression, fused to an IRES GFP sequence (*SF3B1*<sup>WT</sup>-IRES-GFP). When placed in competition, cells infected or not infected with *SF3B1*<sup>WT</sup>-IRES-GFP maintained constant ratios over 10 days (*Figure 2H*), suggesting that short-term expression of *SF3B1* does not alter cellular fitness in either *SF3B1*<sup>neutral</sup> or *SF3B1*<sup>loss</sup> cells. Next, we concomitantly suppressed endogenous *SF3B1* in all cells and expressed *SF3B1*<sup>WT</sup>-IRES-GFP in ~50% of cells. While *SF3B1*<sup>neutral</sup> cells were not affected by partial *SF3B1* suppression, *SF3B1*<sup>loss</sup> cells expressing *shSF3B1* were not compatible with long-term culture. However, *SF3B1*<sup>loss</sup> cells expressing both *shSF3B1* and *SF3B1*<sup>WT</sup>-IRES-GFP persisted (*Figure 2I*), indicating that re-expression of *SF3B1* is sufficient to prevent cell death. Furthermore, *SF3B1*<sup>loss</sup> cells expressing both *shSF3B1* and *SF3B1*<sup>WT</sup>-IRES-GFP had a 20-fold increase in GFP fluorescence, suggesting that the exogenous *SF3B1* construct was more highly expressed in *SF3B1*<sup>loss</sup> cells after suppression of endogenous *SF3B1* (*Figure 2—figure supplement 1C*). Furthermore, stable exogenous *SF3B1* expression is sufficient to restore the proliferation of *SF3B1*<sup>loss</sup>cells expressing shRNAs targeting *SF3B1* (*Figure 2J* and *Figure 2—figure supplement 1D*).

## *SF3B1*<sup>neutral</sup> cells contain excess SF3B1 beyond the requirement for survival

Analyses of *SF3B1* mRNA indicate that *SF3B1*<sup>neutral</sup> cells tolerate partial *SF3B1* suppression because they express more SF3B1 than the minimum amount needed for survival. In both TCGA breast adenocarcinoma data (*Cancer Genome Atlas Network, 2012*) and the Cancer Cell Line Encyclopedia, *SF3B1*<sup>neutral</sup> samples exhibited significantly higher expression of *SF3B1* mRNA relative to *SF3B1*<sup>loss</sup> samples (*Figure 3A* and *Figure 3—figure supplement 1A*; Mann-Whitney $p<10^{-4}$, for both datasets), suggesting excess mRNA over requirements for survival. We validated that *SF3B1*<sup>neutral</sup> breast cancer cell lines (n = 7) express approximately twice as much *SF3B1* mRNA as *SF3B1*<sup>loss</sup> cells (n = 5) by quantitative PCR (*Figure 3B*; $p<10^{-4}$). We also found similar *SF3B1* expression changes between the SF3B1<sup>control-Cal51</sup> and SF3B1<sup>Loss-Cal51</sup> lines (*Figure 3—figure supplement 1B*).

Reductions in *SF3B1* mRNA expression were recapitulated at the protein level. Among breast cancer lines, we found increased SF3B1 protein expression in *SF3B1*<sup>neutral</sup> compared to *SF3B1*<sup>loss</sup> cells (*Figure 3C*) and SF3B1<sup>control-Cal51</sup> vs. SF3B1<sup>Loss-Cal51</sup> cells (*Figure 3D*). We also a found significant linear correlation between SF3B1 mRNA and protein expression (*Figure 3E*, p=0.0018, $R^2 = 0.772$).

These observations suggest that *SF3B1*<sup>neutral</sup> cells tolerate partial *SF3B1* suppression because moderate *SF3B1* suppression leaves them with sufficient residual protein for survival. Indeed, we found detectable SF3B1 levels in *SF3B1*<sup>neutral</sup> cells after *SF3B1* suppression, but failed to detect protein in *SF3B1*<sup>loss</sup> cells after *SF3B1* suppression (*Figure 3F* and *Figure 2—figure supplement 1D*).

A systematic analysis of shRNA-induced mRNA suppression across *SF3B1*<sup>neutral</sup> and *SF3B1*<sup>loss</sup> lines indicated that *SF3B1* mRNA levels can be reduced by ~60% from *SF3B1*<sup>neutral</sup> cell basal levels before proliferation and viability defects are apparent (*Figure 3G*). We suppressed *SF3B1* using shRNAs with different potency to generate a range of *SF3B1* suppression in *SF3B1*<sup>neutral</sup> and *SF3B1*<sup>loss</sup> cells, including shRNAs that suppress SF3B1 by >70% in *SF3B1*<sup>neutral</sup> cells, and determined the impact on

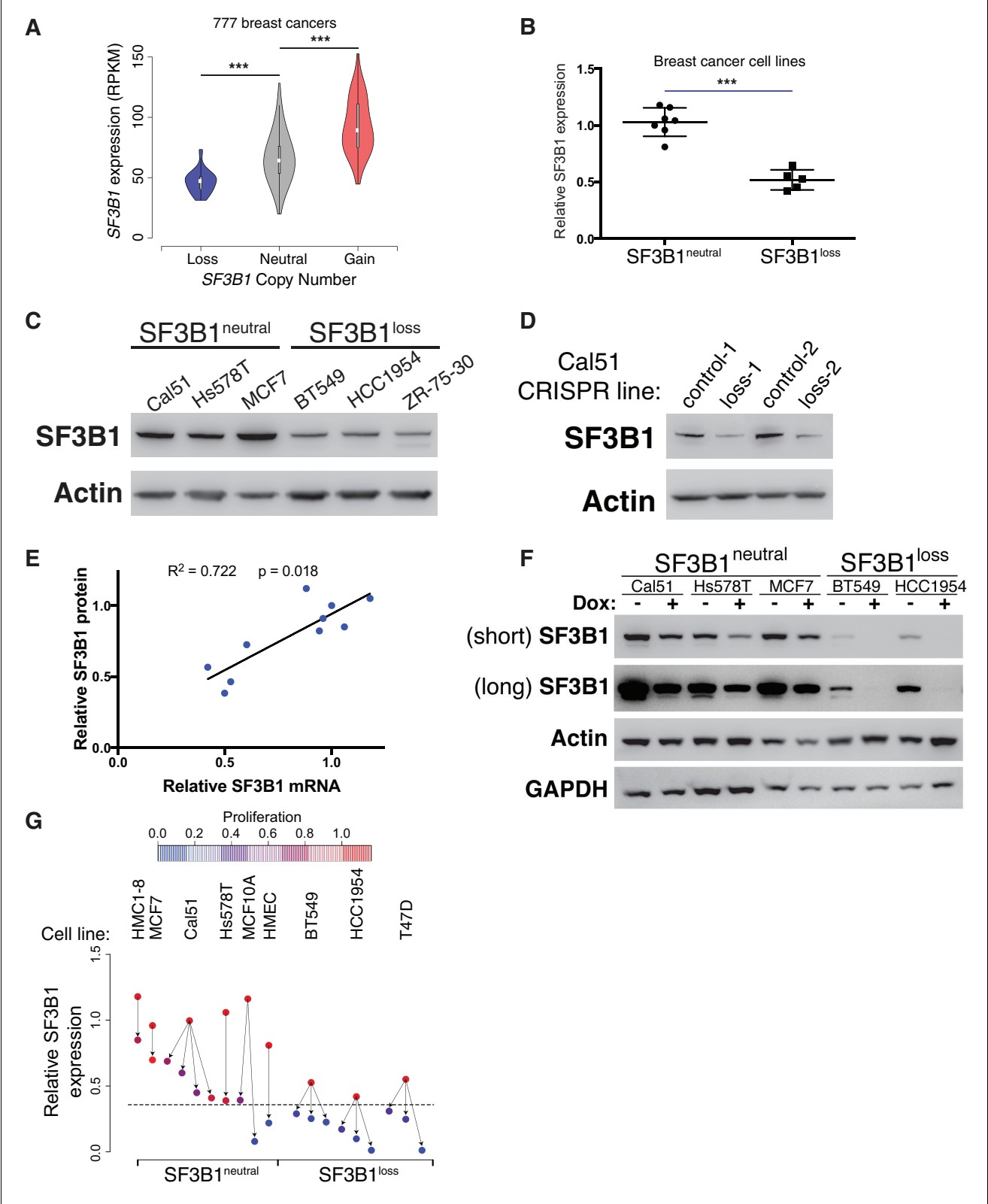

**Figure 3.** *SF3B1^neutral* cells contain excess SF3B1 beyond the requirement for survival. (**A**) *SF3B1* expression from 777 TCGA breast adenocarcinomas segregated by *SF3B1* copy number. Whiskers represent min/max values and boxes represent upper and lower quartile ranges. Width of plots represents relative sample density. (**B**) Quantitative RT-PCR of *SF3B1* expression in breast cancer cell lines. Data points represent individual cell lines, horizontal lines indicate means. (**C**) SF3B1 protein levels in breast cancer cell lines by immunoblot. (**D**) SF3B1 immunoblot from control cells and those

*Figure 3 continued on next page*

*Figure 3 continued*

with single-copy *SF3B1* inactivation by CRISPR. (E) Scatterplot of SF3B1 mRNA and protein expression relative to diploid cell line Cal51 after normalization to actin in a panel of breast cancer cell lines (p=0.0018, $R^2$ = 0.772, regression line slope = 0.789). (F) SF3B1 immunoblot from SF3B1$^{neutral}$ and SF3B1$^{loss}$ cells 4 days after TR-shSF3B1#5 induction by doxycycline. (G) Differences in proliferation 7 days after SF3B1 suppression (per CellTiter-Glo, see Appendix Methods; red=high, blue=low), against the relative level of SF3B1 expression (assessed by qPCR; y-axis) in SF3B1$^{neutral}$ (left) or SF3B1$^{loss}$ (right) cells expressing either shLacZ (origins of arrows) or shSF3B1 (ends of arrows). Origins with multiple arrows represent cell lines subject to more than one SF3B1 shRNA. Each data point represents the mean from at least two replicate experiments. The dashed line represents the estimated minimum threshold of *SF3B1* expression required for survival. For all panels, *p<0.05 **p<0.01 ***p<0.001, and error bars represent ± SD.

The following figure supplement is available for figure 3:

**Figure supplement 1.** Further characterization of SF3B1 as a CYCLOPS gene.

cellular growth seven days after shRNA expression. Although similar reductions in *SF3B1* expression were obtained in *SF3B1$^{neutral}$* and *SF3B1$^{loss}$* lines, the elevated basal levels of *SF3B1* expression in *SF3B1$^{neutral}$* lines enabled them to retain sufficient *SF3B1* for proliferation despite shRNA expression, except in cases when SF3B1 suppression exceeded the 60% threshold of viability.

## *SF3B1* copy-loss selectively reduces the abundance of the SF3b complex

We next asked whether the reduction of SF3B1 protein expression in *SF3B1$^{loss}$* cells preferentially altered specific SF3B1-containing complexes. SF3B1 is a component of the seven-member SF3b sub-complex of the U2 snRNP. Incorporation of SF3b into the U2 snRNP 12S 'core' forms the 15S U2 snRNP, which combines with SF3a to form the mature 17S U2 snRNP (*Figure 4A*) (*Krämer et al., 1999*; *van der Feltz et al., 2012*). We therefore interrogated expression levels of native SF3B1-containing complexes from whole-cell extracts by glycerol gradient sedimentation and gel filtration chromatography. We were able to resolve protein complexes from 29–650 kDa and 650–2,000 kDa using 10–30% glycerol gradients and Sephacryl S-500 gel filtration chromatography, respectively (*Figure 4—figure supplement 1A–B*). This enabled resolution of SF3B1-containing complexes ranging from monomers (155 kDa) to the SF3b sub-complex (450 kDa) to the 15S and 17S U2 snRNPs (790 and 987 kDa, respectively) (*van der Feltz et al., 2012*). We compared these elution profiles between patient-derived and isogenic *SF3B1$^{loss}$* and *SF3B1$^{neutral}$* cells.

We observed significantly lower levels of SF3b in the *SF3B1$^{loss}$* cells. The largest decreases in SF3B1-containing complexes in glycerol gradients were in fractions 4–8, corresponding to ~29–450 kDa (*Figure 4B–F*) and fractions 12–14, corresponding to ~450–650 kDa (*Figure 4G* and *Figure 4—figure supplement 1C*). We saw similar decreases in gel filtration chromatography fractions corresponding to complexes <650 kDa (*Figure 4—figure supplement 1D*). Native western blotting from the pooled glycerol gradient fractions 4–6 indicated the loss of a single SF3B1-containing complex of ~450 kDa (*Figure 4H*). SF3B1 immunoprecipitation from fractions 4–6 resulted in the co-precipitation of SF3b components SF3B3 and SF3B4 in *SF3B1$^{neutral}$* cells, but not of U2 snRNP components SNRPB2 and SF3A3 (*Figure 4—figure supplement 1E*).

Conversely, it appears that U2 snRNP levels are only modestly decreased in *SF3B1$^{loss}$* lines. Levels of SF3B1 in glycerol gradient fraction 25 (corresponding to >650 kDa) were not significantly decreased in *SF3B1$^{loss}$* relative to *SF3B1$^{neutral}$* lines (*Figure 4G* and *Figure 4—figure supplement 1C*). SF3B1 immunoprecipitation from fractions 24–25 resulted in co-precipitation of U2 snRNP components SNRPB2 and SF3A3 (*Figure 4—figure supplement 1F*). U2 snRNA levels are known to track with U2 snRNP levels, and we also did not observe a significant difference in U2 snRNA abundance between *SF3B1$^{neutral}$* and *SF3B1$^{loss}$* lines (*Figure 4I* p=0.35, two-tailed t-test). Similarly, visualization of U2 snRNP complexes using radiolabeled oligonucleotides complementary to the U2 snRNA did not demonstrate differences in 17S U2 snRNP abundance in *SF3B1$^{loss}$* cells (*Figure 4J–K* and *Figure 4—figure supplement 1G*, p=0.68).

These observations suggest that copy-loss of *SF3B1* only modestly affects U2 snRNP abundance but substantially decreases levels of U2 snRNP precursor complexes under steady-state conditions (*Figure 4L*).

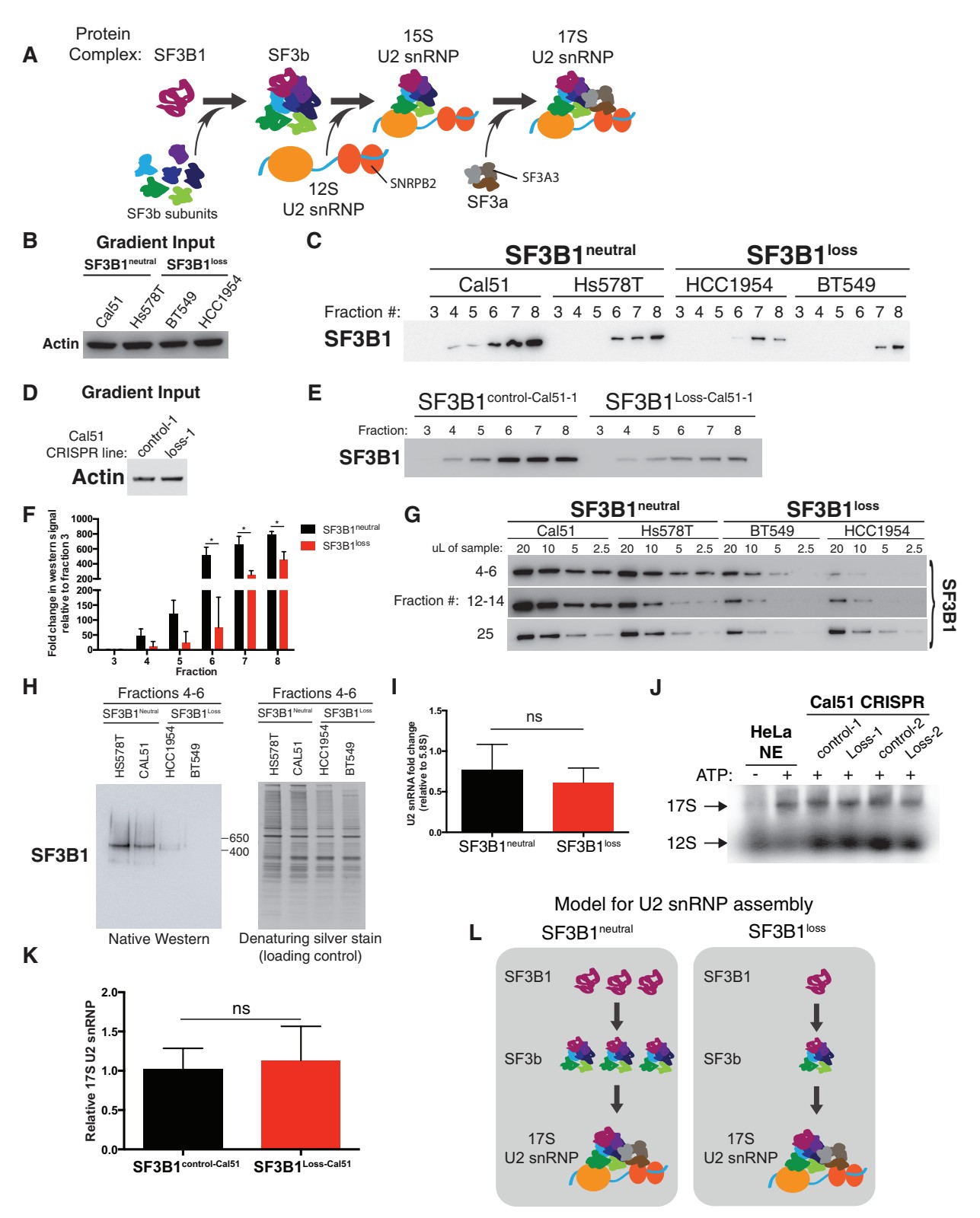

**Figure 4.** *SF3B1* copy-loss selectively reduces the abundance of the SF3b complex. (A) Diagram of U2 snRNP assembly. (B–C) Glycerol gradient fractionation of native whole-cell lysates probed by western blot in breast cancer cell lines and (D–E) isogenic cells generated by CRISPR. (F) Quantification of SF3B1 immunoblots from glycerol gradient fractions 3–8, relative to fraction 3 (n = 3 for each group, see Appendix Methods). (G) Serial dilution of pooled glycerol gradient fractions probed for SF3B1 by immunoblot. (H) (left) SF3B1 Native PAGE immunoblot of pooled glycerol

*Figure 4 continued on next page*

Figure 4 continued

gradient fractions. (right) denaturing silver stain of total protein from the same pooled fractions as loading control. (I) Quantitative RT-PCR for U2 snRNA expression in three *SF3B1*$^{neutral}$ and three *SF3B1*$^{loss}$ breast cancer cell lines. (J) Native agarose gel of U2 snRNP complexes visualized with radiolabeled 2′ O-methyl oligonucleotides complementary to the U2 snRNA. Nuclear extracts were generated from SF3B1$^{control-Cal51}$ and SF3B1$^{Loss-Cal51}$ cells. HeLa cell nuclear extracts (NE) ± ATP were used as controls. Representative image from triplicate experiments. (K) Densitometric quantification of 17S U2 snRNP bands in (J) presented as fold change relative to SF3B1$^{control-Cal51}$ cells. Data are from three replicate experiments. (L) Model for changes to U2 snRNP assembly associated with *SF3B1* copy-loss. For all panels, *$p<0.05$, and error bars represent ± SD, ns = not significant.

The following figure supplement is available for figure 4:

**Figure supplement 1.** SF3B1 copy-loss selectively reduces the abundance of the SF3b complex.

## SF3B1 suppression selectively reduces U2 snRNP abundance in *SF3B1*$^{loss}$ cells

Partial suppression of *SF3B1* leads to substantial reductions of U2 snRNP levels in *SF3B1*$^{loss}$ but not *SF3B1*$^{neutral}$ cells. Although such suppression results in reduced SF3B1 levels in both *SF3B1*$^{loss}$ and *SF3B1*$^{neutral}$ lines, only the *SF3B1*$^{loss}$ lines exhibit concomitant reductions in levels of U2 snRNP components SF3A3 and SNRPB2 (*Figure 5A*). Decreases in SF3A3 and SNRPB2 were observed in glycerol gradient fraction 25, corresponding to the U2 snRNP, most dramatically in *SF3B1*$^{loss}$ lines, and to a lesser extent in one of the *SF3B1*$^{neutral}$ lines (Hs578T; *Figure 5B*). Furthermore, after *SF3B1* suppression, we detected both SF3B1 and SNRPB2 in Sephacryl-S500 fractions containing the >650 kDa protein complexes in *SF3B1*$^{neutral}$ cells but not in *SF3B1*$^{loss}$ cells (*Figure 5D–E* and *Figure 5—figure supplement 1A*). Quantitative PCR also indicated significantly reduced U2 snRNA expression after *SF3B1* partial suppression in *SF3B1*$^{loss}$ cells but not in *SF3B1*$^{neutral}$ cells (*Figure 5F*).

In contrast, suppression of *SF3B1* in *SF3B1*$^{neutral}$ cells appears to substantially decrease levels of SF3b, but not the U2 snRNP. *SF3B1* suppression in *SF3B1*$^{neutral}$ cells only modestly reduced SF3B1 in fraction 25 (*Figure 5B* and *Figure 5—figure supplement 1B*) but instead preferentially reduced SF3B1 from fractions 4–6 (*Figure 5C* and *Figure 5—figure supplement 1C*). Further, no changes in SF3A3 or SNRPB2 expression were observed in total protein from glycerol gradient inputs (*Figure 5A*) or U2 snRNA expression (*Figure 5F*). Therefore, we determined if *SF3B1* suppression in *SF3B1*$^{neutral}$ cells phenocopies the reduced SF3b observed in unperturbed *SF3B1*$^{loss}$ cells. Indeed, *SF3B1*$^{neutral}$ cells with *SF3B1* suppression reduced SF3b levels in glycerol gradient fractions 3–6 approximately to the levels observed in *SF3B1*$^{loss}$ cells (*Figure 5G*). Taken together, these data suggest that the elevated levels of the SF3b sub-complex in *SF3B1*$^{neutral}$ cells relative to *SF3B1*$^{loss}$ cells appear to buffer *SF3B1*$^{neutral}$ cells from reductions in viability following partial *SF3B1* suppression.

Consistent with *Figure 4*, *SF3B1*$^{loss}$ cells without *SF3B1* suppression did not have significantly lower levels of U2 snRNP in fraction 25 than *SF3B1*$^{neutral}$ cells (p=0.16, *Figure 5—figure supplement 1D*), suggesting *SF3B1* copy-loss only reduces levels of the U2 snRNP following further depletion of *SF3B1*.

The relative preservation of the U2 snRNP and larger complexes in *SF3B1*$^{loss}$ cells without *SF3B1* suppression suggests existing SF3b inhibitors might not exploit the specific vulnerability exhibited by *SF3B1*$^{loss}$ cells. SF3b inhibitors modulate U2 snRNP function or subsequent steps during splicing catalysis (*Corrionero et al., 2011*; *Folco et al., 2011*; *Roybal and Jurica, 2010*), thereby altering splicing leading to cell death. With similar U2 snRNP levels in *SF3B1*$^{loss}$ and *SF3B1*$^{neutral}$ cells, the dose at which these effects would be expected to accrue might be similar.

We tested this hypothesis by exposing *SF3B1*$^{neutral}$ cell lines with partial *SF3B1* suppression, and controls to treatment with two SF3b-targeting compounds (Spliceostatin A, and Pladienolide B) and with NSC95397, a compound reported to inhibit splicing activity by an SF3b-independent mechanism (*Berg et al., 2012*). None of these exhibited increased effects on cells with partial *SF3B1* suppression. (*Figure 5—figure supplement 1E–H*). We also evaluated isogenic Cal51 cells with engineered SF3B1 copy-loss and did not observe enhanced sensitivity of SF3B1$^{loss-Cal51}$ cell lines (*Figure 5—figure supplement 1I*).

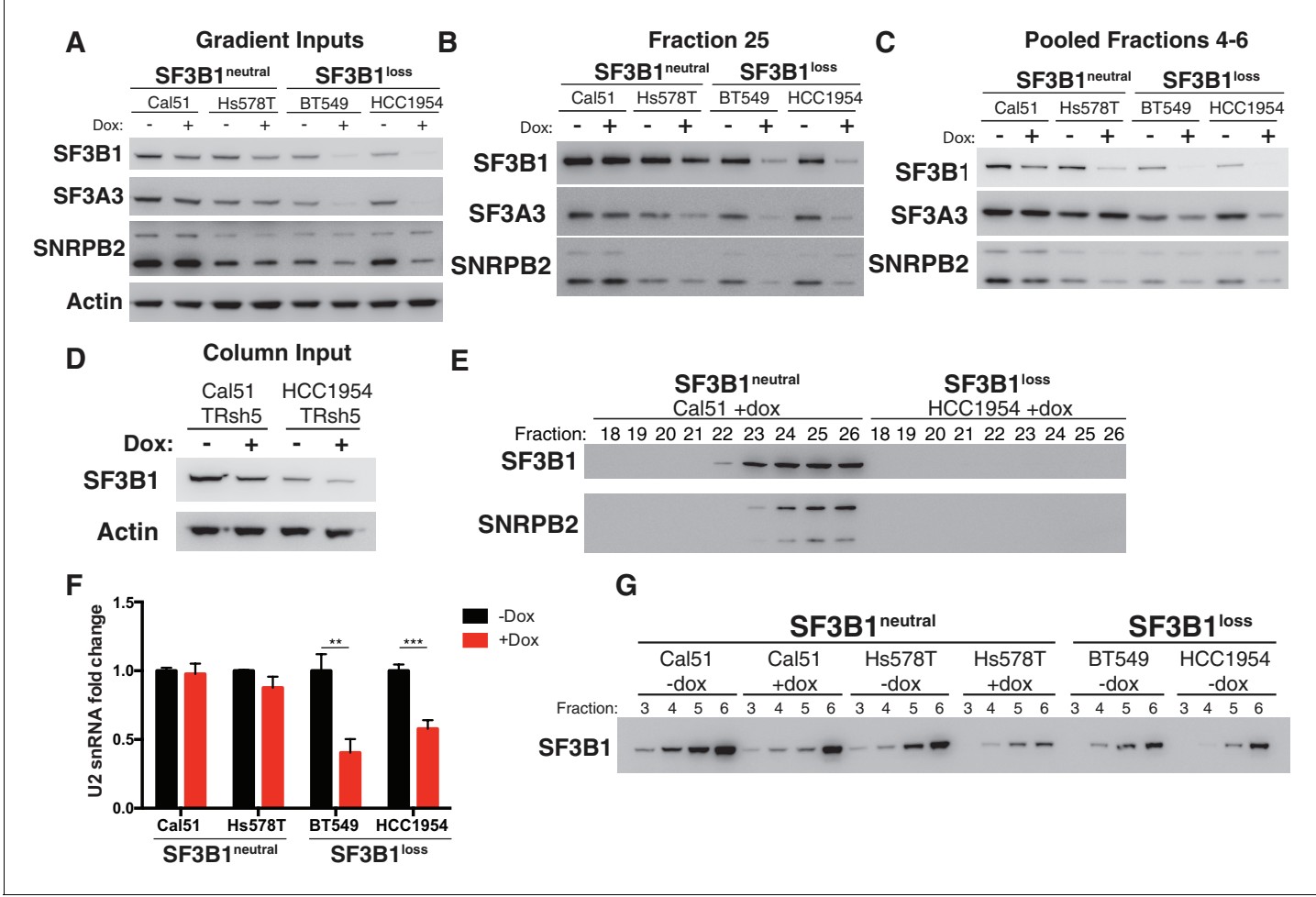

**Figure 5.** Reduced spliceosome precursors and U2 snRNP abundance upon *SF3B1* suppression in *SF3B1^loss^* cells. (**A**) Western immunoblots without and with SF3B1 suppression prior to glycerol gradient fractionation. (**B**) Western immunoblots from glycerol gradient fraction 25 (protein complexes >650 kDa). (**C**) Western immunoblots from pooled glycerol gradient fractions 4–6 (protein complexes ~150–450 kDa). (**D**) Immunoblot from lysates prior to gel filtration chromatography. (**E**) Immunoblot of gel filtration fractions 18–26 (protein complexes >650 kDa) from lysates with *SF3B1* suppression. (**F**) Quantitative RT-PCR for U2 snRNA expression without and with *SF3B1* suppression. (**G**) Glycerol gradient fractions from *SF3B1^neutral^* cells without and with *SF3B1* suppression compared to *SF3B1^loss^* without suppression. For all panels, TR-shSF3B1#5 was used. *p<0.05 **p<0.01 ***p<0.001.

The following figure supplement is available for figure 5:

**Figure supplement 1.** Reduced spliceosome precursors and U2 snRNP abundance upon SF3B1 suppression in SF3B1^loss^ cells.

## Partial *SF3B1* suppression results in splicing defects in *SF3B1^loss^* cells

*SF3B1* is well-established as a splicing factor (*Gozani et al., 1998*; *Zhou et al., 2002*), and intron retention has been reported upon treatment of cells with SF3B1 inhibitors (*Kotake et al., 2007*) while patients harboring *SF3B1* mutations display alterations in alternative splicing (*DeBoever et al., 2015*; *Wang et al., 2011*). We therefore quantified the extent of splicing disruption upon *SF3B1* suppression in *SF3B1^neutral^* and *SF3B1^loss^* cells by RNA sequencing. RNA was isolated 4 days after doxycycline treatment when *SF3B1^loss^* cells arrest in G2/M phase of the cell cycle (*Figure 2F*), but have not initiated apoptosis (*Figure 2G*). We suppressed *SF3B1* in *SF3B1^neutral^* cells to similar or lower levels as seen at steady state in *SF3B1^loss^* cells, and suppressed *SF3B1* in *SF3B1^loss^* cells to even lower levels (*Figure 3F* and *Figure 6—figure supplement 1A–B*). We used juncBase (*Brooks et al., 2011*) and a novel statistical framework to analyze 50,600 splice junctions for intron retention in *SF3B1^neutral^* and *SF3B1^loss^* cells upon *SF3B1* suppression from our RNA sequencing data.

Briefly, we calculated the ratio of percent spliced in (PSI) and spliced out read counts for each splicing junction, but accounted for the probability that any single splicing junction may not be accurately sampled in each cell line using a beta binomial distribution (see Appendix Methods).

All cells showed evidence of increased intron retention following *SF3B1* partial suppression ($p<10^{-5}$), but splicing was significantly more affected in *SF3B1^loss^* cells. Upon *SF3B1* suppression, 7353 transcripts in *SF3B1^loss^* cells showed evidence of significantly (q < 0.1) increased intron retention relative to *SF3B1^neutral^* cells, whereas only 454 transcripts showed evidence of increased intron retention in the reverse direction (*Figure 6A*, $p<10^{-1110}$).

We confirmed increased intron retention in *SF3B1^loss^* cells upon *SF3B1* suppression by qPCR amplifying selected introns of genes from the RNA sequencing analysis (*Figure 6A*). Introns were significantly retained in all seven genes analyzed (*AARS, CALR, DNAJB1, MKNK2, MYH9, RPS8* and *RPS18*), which included *DNAJB1*, a gene previously known to be improperly spliced in cells treated with SF3B1 inhibitors (*Kotake et al., 2007*), and cell essential genes *AARS, RPS8* and *RPS18* (*Figure 6B*).

The SF3b complex is known to regulate 3' splice site selection (*DeBoever et al., 2015*). We therefore analyzed 30,666 alternative 3' splice sites from the RNAseq data in *SF3B1^neutral^* and *SF3B1^loss^* cells. Reduced *SF3B1* expression resulted in significantly more alternative 3' splice site selection in *SF3B1^loss^* cells (353 3' splice junctions in *SF3B1^neutral^* vs. 1540 in *SF3B1^loss^* cells, $p<10^{-121}$, *Figure 6C* and *Figure 6—figure supplement 1C*). We also observed other alterations in splicing. Alternative 5' splice site selection occurred at a significantly higher rate upon *SF3B1* suppression in *SF3B1^loss^* cells (1411 junctions vs 317, $p=9\times10^{-165}$, *Figure 6—figure supplement 1D*). We also observed increased dysregulation of cassette exon inclusion. The proportion of reads including the cassette exon at each junction changed substantially after *SF3B1* suppression in *SF3B1^loss^* cells, but was less substantially altered in *SF3B1^neutral^* cells. (*Figure 6—figure supplement 1E*). We further validated these observations with independent assays for alternative splicing. Specifically, the ratio between alternative long and short isoforms of *MCL1* (that respectively do or do not have anti-apoptotic functions) is known to be regulated by SF3B1 (*Moore et al., 2010*). After *SF3B1* suppression, this ratio was significantly biased towards the short isoform in *SF3B1^loss^* cells relative to *SF3B1^neutral^* cells (*Figure 6D–E*). Together, these data indicate that *SF3B1* suppression more substantially dysregulates splicing of the transcriptome of *SF3B1^loss^* cells.

Spliceosome components, including SF3B1, are thought to assemble and function in sub-nuclear compartments known as nuclear speckles (*Spector and Lamond, 2011*). Inhibition of splicing or transcription has been shown to induce formation of enlarged 'mega-speckles' (*Kotake et al., 2007*; *Loyer et al., 1998*). We therefore performed an unbiased quantification of the number and size of SC-35$^+$ speckles per nucleus using a custom image analysis pipeline with CellProfiler software (*Kamentsky et al., 2011*).

*SF3B1^neutral^* cells did not display changes in SC-35+ speckles after *SF3B1* partial suppression, but *SF3B1^loss^* nuclei contained significantly fewer speckles and increased speckle area (*Figure 6F–H*). The presence of defective alternative splicing, intron retention and formation of mega-speckles uniquely in *SF3B1^loss^* cells after partial *SF3B1* suppression further supports the gross defects in splicing observed by RNA sequencing.

Moreover, upon *SF3B1* suppression, 513 genes were differentially expressed at an FDR < 10% in *SF3B1^loss^* cells and only 306 genes were differentially expressed in *SF3B1^neutral^* cells (*Supplementary file 1E*, $p<10^{-4}$, Fischer's exact test). Gene set enrichment analysis revealed 24 KEGG pathways significantly enriched in *SF3B1^loss^* cells and only nine pathways altered in *SF3B1^neutral^* cells (*Figure 6I* and *Supplementary file 1F*). We also identified a statistically significant core set of 89 genes differentially expressed across all cell lines upon *SF3B1* suppression regardless of *SF3B1* copy number ($p<10^{-4}$).

Upon *SF3B1* expression, 348 genes were differentially expressed in *SF3B1^loss^* cells and 393 genes were differentially expressed in *SF3B1^neutral^* cells, an insignificant difference (p=0.07, *Supplementary file 1G*). We observed more differentially expressed genes in *SF3B1^loss^* cells from *SF3B1* suppression than from *SF3B1* expression (p<0.0001), however, there was no difference between the effects of *SF3B1* suppression and expression in *SF3B1^neutral^* lines (p=0.7). Taken together, these data indicate that partial *SF3B1* suppression more severely impacts the transcriptome of *SF3B1^loss^* cells.

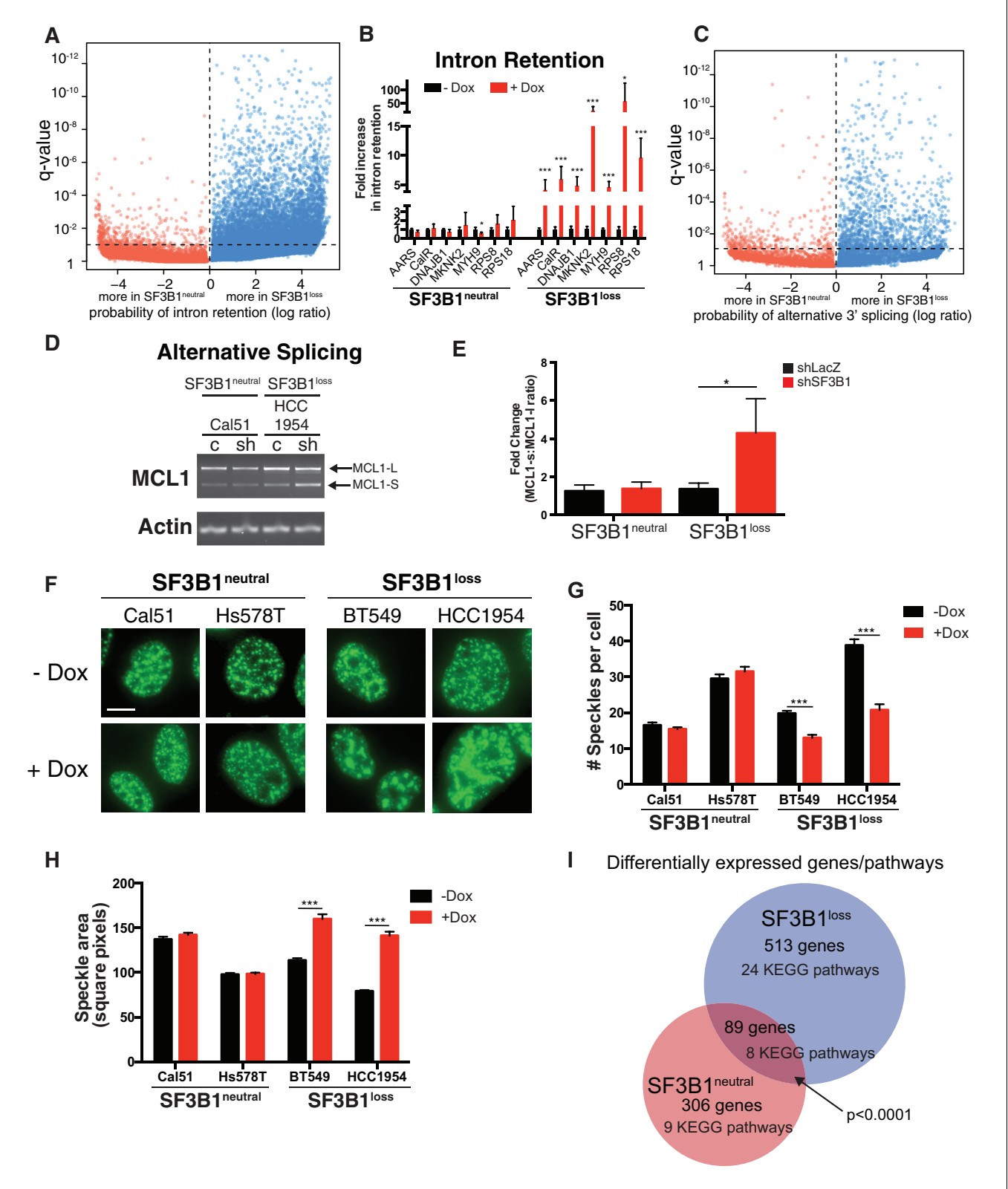

**Figure 6.** Partial *SF3B1* suppression results in splicing defects in *SF3B1*<sup>loss</sup> cells. (A) Statistical significance of intron retention (see Materials and methods) across all exon-intron junctions (dots) in *SF3B1*<sup>neutral</sup> (red) and *SF3B1*<sup>loss</sup> cells (blue) after *SF3B1* suppression. The horizontal dashed line represents the significance threshold (q < 0.01) and the vertical dashed line segregates intron-exon junctions more likely to be altered in *SF3B1*<sup>neutral</sup> (left) or *SF3B1*<sup>loss</sup> cells (right). (B) qPCR for a single intron within the indicated gene without and with *shSF3B1* induction by doxycycline

*Figure 6 continued on next page*

*Figure 6 continued*

(*SF3B1*[neutral] n = 3, *SF3B1*[loss] n = 3, averaged from TR-shSF3B1#3 and TRshSF3B1#5. SF3B1[control-Cal51] and SF3B1[Loss-Cal51] n = 2 each, averaged from TR-shSF3B1#3). (C) Statistical significance of alternative 3' splice site selection (see Materials and methods) across 3' splice junctions (dots) in *SF3B1*[neutral] (red) and *SF3B1*[loss] cells (blue) after *SF3B1* suppression (as in panel A). (D) Representative RT-PCR from SF3B1[neutral] and SF3B1[loss] cells after *SF3B1* knockdown. 'c' represents LacZ control hairpins, 'sh' represents shSF3B1#4. Arrows represent product sizes for MCL-L and MCL-S. (E) Densitometric quantification of the ratio of MCL1-S:MCL1-L, relative to shLacZ-expressing controls (mean ± SD, n = 3 per group averaged from shSF3B1#3, and #4). (F) Immunofluorescent images of nuclear speckles by anti-SC35 (SRSF2) staining. Scale bar = 5 uM. (G) Quantification of number of nuclear speckles and (H) speckle area per cell across at least 100 nuclei. For (F–H) TRshSF3B1#5 was used. (I) Number of differentially expressed genes upon *SF3B1* suppression (q < 0.1) and the number of enriched KEGG pathways amongst indicated gene set (q < 0.05). For all panels, *p<0.05 **p<0.01 ***p<0.001.

The following figure supplement is available for figure 6:

**Figure supplement 1.** Partial SF3B1 suppression results in splicing defects in SF3B1[loss] cells.

## Suppression of *SF3B1* reduces tumor growth in *SF3B1*[loss] xenografts

To test the effects of *SF3B1* partial suppression in vivo, we generated xenografts from luciferase-labeled SF3B1[Loss-Cal51-1], SF3B1[control-Cal51-1] cells and naturally occurring *SF3B1*[neutral] and *SF3B1*[loss] cells (Cal51 and HCC1954, respectively) all containing TR-shSF3B1#3. Interventional studies were performed where animals were placed on doxycycline upon detection of palpable tumors and validated for *SF3B1* suppression (*Figure 7—figure supplement 1A*). SF3B1[Loss-Cal51-1] and SF3B1[control-Cal51-1] cells generated tumors of similar volume in the absence of doxycycline (*Figure 7A*; p=0.7, repeated measures ANOVA). However, partial suppression of *SF3B1* reduced the growth of (*Figure 7B–C*) and number of proliferative Ki67+ cells in (*Figure 7D–E*) xenografts from SF3B1[Loss-Cal51-1] cells but not SF3B1[control-Cal51-1] cells (p=0.001 for both assays). Similarly, reduced tumor growth was observed in naturally occurring *SF3B1*[loss] HCC1954 xenografts and not in *SF3B1*[neutral] Cal51 xenografts (*Figure 7F*).

## Compounds that induce degradation of SF3B1 selectively kill SF3B1[loss] cells

The lack of efficacy of small molecule splicing modulators targeting SF3B1 prompted us to evaluate other compounds that could pharmacologically target the SF3B1 CYCLOPS dependency. Deubiquitinase inhibitors (DUBi's) are a class of compounds that increase protein degradation by preventing the removal of ubiquitin from substrates, thereby enhancing target protein degradation by the proteasome. We first sought evidence for SF3B1 ubiquitination by treating cells with the proteasome inhibitor, epoxomicin, to increase the accumulation of ubiquitinated proteins. Immunoprecipitation of SF3B1 in denaturing conditions after epoxomicin treatment resulted in the detection of ubiquitinated SF3B1 (*Figure 8—figure supplement 1A–B*). Furthermore, proteasome inhibition by MG-132 increased SF3B1 protein levels, suggesting SF3B1 is regulated by the proteasome (*Figure 8—figure supplement 1C*).

To determine if DUBi's could decrease SF3B1 protein expression, we evaluated seven candidate DUBi's for reduced SF3B1 expression by western blot (data not shown) and identified three DUBi's with the ability to reduce SF3B1 protein levels within 24 hr (*Figure 8A*). Of the 3 DUBi's that enhance SF3B1 degradation, one semi-selective DUBi b-AP15 (*D'Arcy et al., 2015*), demonstrated enhanced sensitivity in *SF3B1*[loss] cells, and also cells with reduced *SF3B1* expression by RNAi (*Figure 8B*). The enhanced sensitivity in *SF3B1*[loss] cells or cells with reduced *SF3B1* expression by RNAi was not observed upon treatment with SJB3-019A or the pan-DUBi, PR-619 (*Figure 8—figure supplement 1D*), suggesting inhibition of a specific DUB enzyme may mediate this effect. SF3B1[Loss-Cal51] cells also had more Annexin V apoptotic cells than SF3B1[control-Cal51] cells when treated with nanomolar concentrations of b-AP15 (*Figure 8C*). Immunoblots for SF3B1 after b-AP15 treatment revealed decreased SF3B1 only occurred in SF3B1[Loss-Cal51-2] cells (*Figure 8D* and *Figure 8—figure supplement 1E*). These data suggest the enhanced sensitivity may be due, in part, to reduction of SF3B1 in SF3B1[Loss-Cal51] cells.

Treatment of cells with b-AP15 resulted in splicing alterations preferentially in *SF3B1*[loss] cells. We determined the effect of b-AP15 treatment on previously identified splicing alterations that result

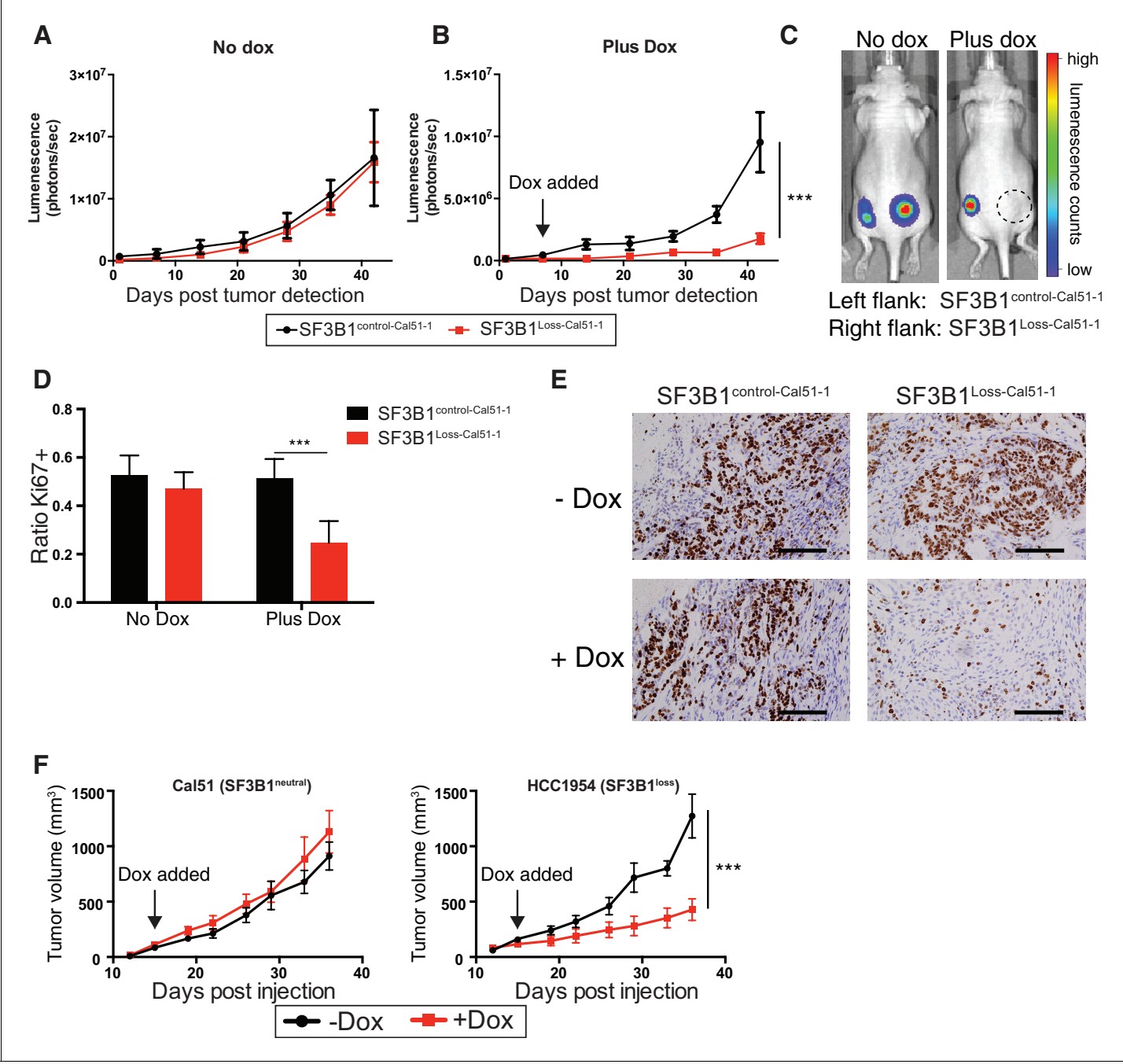

**Figure 7.** SF3B1 suppression inhibits tumor growth in vivo. Luminescent quantification of xenograft growth from SF3B1$^{control-Cal51-1}$ (black) and SF3B1$^{Loss-Cal51-1}$ (red) tumors (**A**) without doxycycline (n = 4) and (**B**) with doxycycline (n = 17) using TR-shSF3B1 #3. (**C**) Representative animal images overlaid with heat maps from bioluminescent tumor detection. Dashed circle represents region where established tumor was detected prior to doxycycline treatment. (**D**) Quantification of Ki67+ cells from xenografts 42 days post tumor detection using CellProfiler ≥ 2440 nuclei were scored for each tumor, ≥ 3 tumors per group (see Materials and methods). (**E**) Representative Ki67 immunohistochemistry images of xenografts quantified in panel (**D**). (**F**) Growth of established tumors for Cal51 and HCC1954 xenografts without doxycycline (black) or with doxycycline (red) using TR-shSF3B1 #3. For Cal51 and HCC1954 no doxycycline groups (n = 13); plus doxycycline groups (n = 12). For all panels, ***p≤0.001.

The following figure supplement is available for figure 7:

**Figure supplement 1.** SF3B1 suppression inhibits tumor growth in vivo.

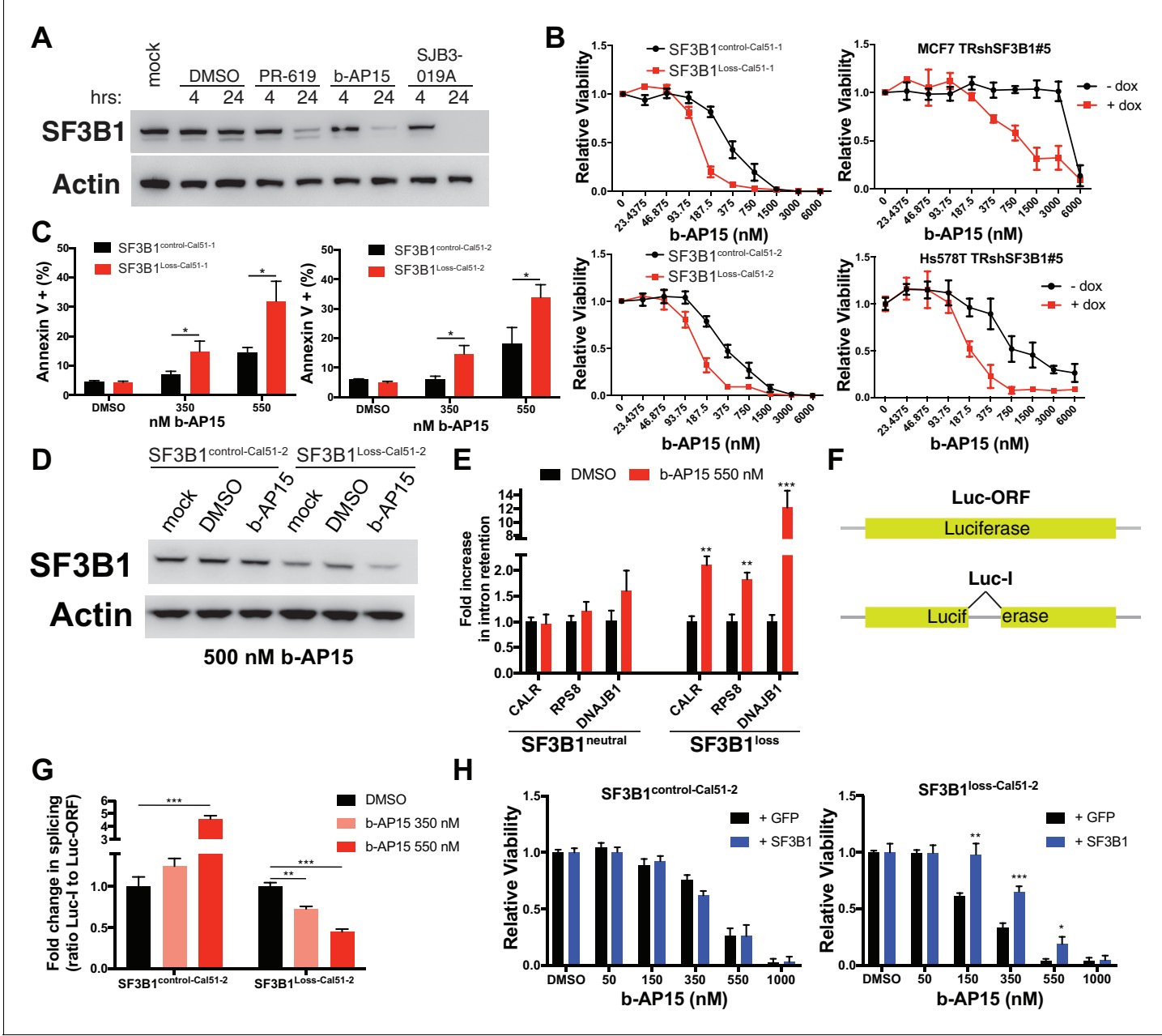

**Figure 8.** Deubiquitinase inhibitor b-AP15 can induce SF3B1 degradation and selectively kill SF3B1[loss] cells. (A) SF3B1 immunoblot after DUBi treatment of the SF3B1[neutral] cell line, Cal51, at the indicated time points. (B) b-AP15 dose response curves in isogenic cell contexts. SF3B1[control-Cal51] and SF3B1[Loss-Cal51] or SF3B1[neutral] cells with or without doxycycline expressing TR-shSF3B1 #5, to phenocopy the reduced SF3B1 expression observed in SF3B1[loss] cells, were assayed 48 hr after b-AP15 treatment. (C) Percent of Annexin V positive cells 24 hr after b-AP15 treatment in SF3B1[control-Cal51] and SF3B1[Loss-Cal51] cells. (D) SF3B1 immunoblot 48 hr after treatment with 500 nM b-AP15 in SF3B1[control-Cal51-2] and SF3B1[Loss-Cal51-2] cells. (E) qPCR for a single intron within the indicated gene with 550 nM b-AP15 or DMSO control (*SF3B1[neutral]* n = 2, *SF3B1[loss]* n = 2, averaged from SF3B1[control-Cal51] and SF3B1[Loss-Cal51] cells. (F) Schematic of luciferase reporter constructs used in (G). Yellow rectangles represent luciferase protein coding regions. Angled black line represents the location of the intron. (G) Fold change in luciferase splicing reporter signal after b-AP15 treatment. Data represented as the ratio of spliced luciferase signal (Luc-I) to luciferase ORF signal (Luc-ORF) relative to their respective DMSO luciferase signal. DMSO treatment (black), 350 nM b-AP15 (light red) and 550 nM b–AP15 (dark red). (H) b-AP15 dose response in *SF3B1[control-Cal51-2]* and *SF3B1[Loss-Cal51-2]* cells expressing GFP (black) or SF3B1 (blue). For all panels, *p<0.05 **p<0.01 ***p<0.001.

The following figure supplement is available for figure 8:

**Figure supplement 1.** Deubiquitinase inhibitor b-AP15 can induce SF3B1 degradation and selectively kill SF3B1[loss] cells.

from SF3B1 suppression. *SF3B1^loss* cells but not *SF3B1^neutral* cells exhibited increased intron retention in *CALR, RPS8 and DNAJB1* after treatment with 550 nM b-AP15 (*Figure 8E*). We also generated stable luciferase expressing cell lines with either a luciferase gene interrupted by a chimeric globin/immunoglobulin intron (Luc-I), or a luciferase open reading frame that does not require splicing as a control (Luc-ORF) (*Figure 8F*) (*Younis et al., 2010*). b-AP15 treatment resulted in a dose-dependent decrease in the ratio of Luc-I:Luc-ORF in *SF3B1^loss-Cal51-2* cells, but not *SF3B1^control-Cal51-2* cells, suggesting b-AP15 induced SF3B1 degradation may mediate splicing reporter loss of function (*Figure 8G*). In contrast, b-AP15 also enhanced *SF3B1^control-Cal51-2* splicing reporter signal. Lastly, rescue experiments in which we expressed exogenous *SF3B1* increased the concentration of b-AP15 required to impact cell viability in *SF3B1^loss* cells, but did not affect the sensitivity of *SF3B1^control-Cal51-2* cells (*Figure 8H*).

Taken together, these results suggest that the DUBi, b-AP15, can decrease SF3B1 stability with increased sensitivity in *SF3B1^loss* cells. They also suggest DUBi's as a potential class of small molecules to be used as a general approach to target candidate CYCLOPS gene dependencies identified here.

## Discussion

Our analysis indicates that CYCLOPS vulnerabilities are the most enriched class of copy-number associated cancer vulnerabilities, and a focused analysis of this class of vulnerabilities identified 124 candidate CYCLOPS genes. In previous work involving 86 cell lines (*Nijhawan et al., 2012*), we had identified 55 candidate CYCLOPS genes at a q < 0.25 significance threshold. Here, we evaluated over twice as many cell lines (179), generating more than twice as many candidate CYCLOPS genes (124) at a more stringent significance threshold (q < 0.1). We also identified trans gene dependencies associated with copy gain, but the regions of copy gain are often large and contain many genes. As a result, it is difficult to assign a specific gene within the gained region as causing the dependency when gained. Nonetheless these classes of copy number associated gene dependencies warrant further investigation.

Our enhanced ability to detect candidate CYCLOPS genes with larger sample sizes suggests that analysis of additional lines is likely to reveal more candidates. For example, we were unable to identify tumor-type specific CYCLOPS genes due to the relatively small numbers of cell lines screened in each cell lineage. In addition, detection of CYCLOPS vulnerabilities requires partial gene suppression, in which the optimal level of suppression may vary by gene. Among SF3B1 shRNAs, ~50% suppression was achieved. We do not know the extent to which the shRNAs used in Project Achilles matched the required levels to expose CYCLOPS vulnerabilities across other genes. These limitations suggest expanding RNAi viability screens to include more cancer cell lines and shRNAs may yield further CYCLOPS dependencies.

The candidate CYCLOPS genes we identified shared the following features: (1) consistent expression across normal tissues, (2) altered expression when affected by copy-number alterations suggesting lack of feedback regulation, (3) location in genomic regions that frequently undergo copy-loss. These features may predispose genes towards meeting CYCLOPS criteria. However, we cannot exclude the possibility that CYCLOPS genes with different features exist but were not identified due to the limitations of the screening data or analyses.

Components of the spliceosome were the most frequently represented genes among CYCLOPS candidates, with 20 representatives, including *SF3B1*. The spliceosome, which is essential for cell survival, has previously been identified as a therapeutic target in cancer (*Kaida et al., 2007*; *Kotake et al., 2007*; *Mizui et al., 2004*; *Webb et al., 2013*). Also, the magnitude of *SF3B1* copy-loss required to unveil the SF3B1 CYCLOPS dependency (e.g. single copy loss in a tetraploid cell) will require further study. Our work further substantiates the importance of the spliceosome as a therapeutic target in cancer, and suggests CYCLOPS vulnerabilities as a possible mechanism of dependency.

Mutation of the spliceosome components *SF3B1* and *U2AF1* are recurrent driver events in many cancer types including chronic lymphoblastic leukemia, myelodysplastic syndrome, uveal melanoma and breast cancer (*Ellis et al., 2012*; *Harbour et al., 2013*; *Imielinski et al., 2012*; *Papaemmanuil et al., 2011*; *Wang et al., 2011*; *Yoshida et al., 2011*). Recent evidence demonstrated that mutations in SF3B1 and SRSF2 can confer preferential sensitivity to chemical modulation

of the spliceosome (*Lee et al., 2016*; *Obeng et al., 2016*). In contrast, we observed partial copy-loss of *SF3B1* and *U2AF1*, likely the result of passenger copy-number alterations, resulted in CYCLOPS vulnerabilities. Consistent with our observations, recent studies of *Sf3b1*$^{+/-}$ mice suggest that partial *SF3B1* loss does not generate haploinsufficient phenotypes (*Matsunawa et al., 2014*; *Wang et al., 2014*) in contrast to previous data (*Visconte et al., 2012*). Our observation that ESS1, a *SF3B1*$^{K666N}$ mutant cell line, was sensitive to partial SF3B1 suppression is consistent with a prior report that SF3B1-mutant cancer cells were found to depend on remaining wild-type copy (*Zhou et al., 2015*). Mutant SF3B1 is also associated with aberrant 3' splice site selection and ~50% rate of nonsense mediated decay in processed transcripts, suggesting that those transcripts exhibit partial loss of function (*Darman et al., 2015*). Taken together, *SF3B1* mutations may be associated with CYCLOPS dependencies, at least in some cases. Therefore, it is possible that similar approaches to targeting the spliceosome can be exploited in *SF3B1*-mutant and *SF3B1*$^{loss}$ cancers. However, it is also possible that alternative strategies will be required for *SF3B1*-mutant and *SF3B1*$^{loss}$ cancers, due to their different effects on spliceosome assembly and activity.

We found that partial *SF3B1* suppression is tolerated in *SF3B1*$^{neutral}$ cells due to a stable pool of SF3b sub-complex outside of the U2 snRNP. To our knowledge, this is the first observation of such a pool of SF3b, and raises the possibility that SF3b has functions in addition to its role in the U2 snRNP. Precedence exists for such functions as SF3B1 is reported to be a component of the poly-comb repressor complex and also directly associates with nucleosomes (*Isono et al., 2005*; *Kfir et al., 2015*). We also observed decreased protein levels of SNRPB2, SF3A3 and the U2 snRNA in *SF3B1*$^{loss}$ cells after *SF3B1* suppression (**Figure 5A and F**). It is possible that due to the decreased amount of assembled 17S U2 snRNP, the U2 precursor complexes are less stable and degraded to maintain proper U2 snRNP stoichiometry in *SF3B1*$^{loss}$ cells. Although these experiments were performed prior to detection of apoptosis, we cannot exclude the possibility that decreased U2 precursor complexes are a result of decreased cell viability beyond the level of detection.

Small molecule inhibitors of SF3B1, and the spliceosome in general, are an emerging class of anti-neoplastic agents. The finding that many candidate CYCLOPS genes are splicing factors suggests further efforts should be paid to identifying CYCLOPS vulnerabilities that can be exploited by spliceosome inhibitors.

In the case of *SF3B1*, we find that *SF3B1* copy-loss reduces the levels of the SF3b complex, but not assembled U2 snRNP, which suggests that inhibitors of the assembled U2 snRNP will not adequately distinguish between *SF3B1*$^{loss}$ and *SF3B1*$^{neutral}$ cells. This includes most current inhibitors of SF3B1 (*Corrionero et al., 2011*; *Folco et al., 2011*; *Roybal and Jurica, 2010*). Understanding which SF3B1 complexes are bound by these compounds, and how they affect the stability, assembly, or function of the U2 snRNP, will be important to glean mechanistic insight into the landscape of small molecule modulation of SF3B1.

More generally, many CYCLOPS vulnerabilities are unlikely to be exploited by compounds that directly inhibit CYCLOPS gene enzymatic activity. Most of the candidate CYCLOPS genes we identified are not enzymes. Many of them are components of multi-protein complexes. One approach to exploiting these vulnerabilities may be to disrupt the incorporation of the proteins encoded by such CYCLOPS genes into these larger complexes. Another approach may be to reduce levels of protein encoded by CYCLOPS genes, either by interfering with the transcription of these genes (*Arrowsmith et al., 2012*; *Kwiatkowski et al., 2014*) or by enhancing protein degradation (*Bondeson et al., 2015*; *D'Arcy et al., 2015*; *Winter et al., 2015*). An advantage to CYCLOPS targets such as SF3B1 is that only moderate (~50%) reductions in protein levels need to be achieved to take advantage of the vulnerability.

Finally, we evaluated DUBi's as a potential small molecule to target the SF3B1 CYCLOPS vulnerability. We found that the DUBi b-AP15 could reduce SF3B1 protein levels and preferentially target *SF3B1*$^{loss}$ cells. A derivative of b-AP15, VLX1570, is already in use in humans (Trial ID: NCT02372240). Our results indicate SF3B1 copy number should be tested as a potential biomarker of response for patients treated with this compound.

Although b-AP15 specifically suppressed growth of cells with loss of SF3B1, it is possible other DUBi's would be even more specific. There are approximately 90 DUBs in humans (*Nijman et al., 2005*), and the DUBi's we tested are relatively non-selective. Further investigation of the specific DUBs that support SF3B1 protein expression would enable identification DUBi's whose effects are more targeted to the SF3B1 dependency.

The finding that a DUBi specifically suppresses growth of cells with loss of SF3B1 also supports the use of DUBi's to target any of the 124 candidate CYCLOPS genes we identified, representing a potential paradigm for treating cancers based upon non-oncogenic genetic events.

## Materials and methods

### Analysis of genome-wide copy-number associated cancer dependencies

Gene-level relative copy-numbers were downloaded from the CCLE portal (http://www.broadinstitute.org/ccle, data version 4/06/2012) totaling 23,124 genes. Gene-level dependencies were obtained for 214 cell lines from Project Achilles (version 2.4.3). Of the 214 cell lines, only 179 had corresponding copy-number data and were used for subsequent analyses. ATARiS gene dependency scores were used to estimate the effect of shRNA-induced gene suppression on cell viability (*Shao et al., 2013*) totaling 8724 gene dependencies. Pearson correlation coefficients and associated p-values were calculated for the association of viability after suppression of each gene with the copy number of all genes. P-values were corrected for multiple hypotheses using the Benjamini-Hochberg method (*Benjamini and Hochberg, 1995*). We considered associations between copy-numbers of every gene in the genome and dependencies of every gene with ATARiS scores. Large copy-number events affecting many neighboring genes often generated identical significant copy-number:gene dependency associations for copy-numbers associated with multiple genes. We considered these to reflect a single gene whose copy-number was responsible for the association. When the gene dependency was one of the genes contained within the copy-number altered region, we nominated that gene as the source of the association. Likewise, if a gene dependency reflected a paralog of a gene within the copy-number altered region, we nominated its paralog as the dependency-associated gene within the altered region.

### CYCLOPS analysis

We determined the significance of differences in ATARiS scores between copy-neutral and –loss lines for every gene by comparing the observed data to data representing random permutations of copy-number class labels, each maintaining the number of cell lines and lineage distribution in each class. Copy-number classes were assigned as copy-loss for cells with $\log_2$ relative copy number ratios $\leq -0.35$ and copy-neutral otherwise. Genes for which fewer than two cell lines exhibited copy loss were excluded from the analysis.

### Generation of heterozygous $SF3B1^{loss}$ cells by CRISPR-Cas9

For SF3B1$^{Loss-Cal51-1}$ cells, sgRNAs targeting the first constitutively expressed coding exon of *SF3B1* (exon 2) were designed with the aid of a web-based application (http://crispr.mit.edu/). Sense and anti-sense oligonucelotides were annealed and cloned into *BbsI* site of pX458 (Addgene) and verified by Sanger sequencing. Single GFP+ cells were sorted by FACS and plated at low density for single cell cloning. SF3B1$^{control-Cal51-1}$ cells were processed identically, but did not have inactivating *SF3B1* mutations. (See supplementary information for further details.)

For SF3B1$^{Loss-Cal51-2}$ cells, a Cas9 construct co-expressing two sgRNAs and GFP was used to delete a 57 kb region encoding *SF3B1*. The guide RNA targeting the 5' upstream of *SF3B1* used a mismatch from a heterozygous SNP (rs3849362) in Cal51 to bias towards mono-allelic deletion of *SF3B1*. Single GFP+ cells were plated as described above and expanded. One of these was validated by PCR to harbor a 57 kb deletion encoding SF3B1. This was designated 'SF3B1$^{Loss-Cal51-2}$' for subsequent experiments. Another one of these was found by PCR not to harbor this deletion and was designated as the control cell line for subsequent experiments ('SF3B1$^{control-Cal51-2}$').

### Glycerol gradient sedimentation

Glycerol gradient sedimentation was performed as previously described (*Hartmuth et al., 2012*) with slight modifications for use with whole-cell lysates. Briefly, 10–30% glycerol gradients were formed by layering 10% glycerol gradient buffer (20 mM Hepes-KOH (pH 7.9), 150 mM NaCl, 1.5 mM MgCl$_2$10% glycerol) on top of a 30% glycerol buffer with identical salt concentrations. Gradients were formed using a Gradient Station (Biocomp Instruments). Cells were lysed in 'IP lysis buffer' (50 mM Tris, 150 mM NaCl and 1% Triton X-100). 400 uL containing 1–3 mg of crude lysate was loaded

per gradient in SW55 centrifuge tubes and spun at 55,000 RPM for 3.5 hr at 4C. A total of 25 200 uL fractions were collected by manually pipetting from the top of the gradient. Recombinant proteins of known mass were run in parallel gradients as controls.

## Gel filtration chromatography

Sephacryl S-500 (17-0613–05, GE Healthcare) columns were packed into a 50 × 1.5 cm column and equilibrated with column buffer (10 mM Tris, 60 mM KCl, 25 mM EDTA, 1% Triton X-100 and 0.1% sodium azide). Whole-cell lysates were collected in IP lysis buffer as described above and incubated with 0.5 mM ATP, 3.2 mM $MgCl_2$ and 20 mM creatine phosphate (di-Tris salt) for 20 min at 30C to dissociate multi-snRNP spliceosomal complexes. 2 mL of lysate containing 5 mg of protein was loaded on columns and 90 1.5 mL fractions were collected overnight at 4C.

## RNA sequencing analysis

Total RNA was extracted and treated by DNAse digestion. RNA quality was determined by bioanalyzer (Agilent) and samples with RIN values >7 were used for sequencing. mRNAs were enriched with the NEBNext Poly(A) mRNA Magnetic Isolation Module (New England BioLabs, #E7490S) and library preparations were performed with the NEBNext Ultra Directional RNA Library Prep Kit (New England BioLabs, #E7420S). 75 bp paired reads were generated using a NextSeq 500 sequencer (Illumina). Approximately 50 million reads per sample were generated. FASTQ files were aligned using TOPHAT v1.4 with parameters '–mate-inner- dist 300 –mate-std-dev 500 –no-sort-bam –no-convert-bam -p 4'. JuncBase was used to identify read counts at splice junctions (see Appendix Methods).

## Generation of xenografts and growth assessment

All animal husbandry was done with the approval of the Dana-Farber Cancer Institute IACUC. 1 × $10^6$ SF3B1[control-Cal51-1] or SF3B1[Loss-Cal51-1] cells expressing TR-shSF3B1 #3 were subcutaneously injected into opposing flanks of nude mice (Foxn1 [nu/nu], Harlan). Animals were randomized to control group or doxycycline treatment after detection of a palpable tumor on either flank. Mice in the doxycycline treatment arm were continuously fed a doxycycline diet (2000 ppm). Mice were sacrificed at the end of the experiment, or when endpoints were reached based on failure to thrive according to IACUC recommendation. Repeated measures two-way ANOVA was used to assess significance.

## Acknowledgements

We would like to thank Dr. Minoru Yoshida for supplying Spliceostatin A, and Azin Sayad and Benjamin Neel for sharing extended data from their shRNA gene dependency screen (*Marcotte et al., 2016*). The RNA sequencing data have been deposited in NCBI's Gene Expression Omnibus (accession number GSE81978). This work was supported in part by NIH grants R01 CA188228 (RB), U01 CA176058 (WCH), F32 CA180653 (BRP), F30 CA192725 (WJG), and R01 GM043375 (RR), the Sontag, Gray Matters, and Pediatric Low-Grade Astrocytoma Foundations (RB), and the Friends for Life Fellowship (BRP).

## Additional information

### Funding

| Funder | Grant reference number | Author |
| --- | --- | --- |
| National Cancer Institute | F32 CA180653 | Brenton R Paolella |
| Sontag Foundation | | Rameen Beroukhim |
| The Grey Matters Foundation | | Rameen Beroukhim |
| The Pediatric Low Grade Astrocytoma Foundation | | Pratiti Bandopadhayay<br>Charles D Stiles<br>Rameen Beroukhim |
| Friends for Life Fellowship | | Brenton R Paolella |

| National Cancer Institute | R01 CA188228 | Rameen Beroukhim |
| National Cancer Institute | U01 CA176058 | William C Hahn |
| National Cancer Institute | F30 CA192725 | William J Gibson |
| National Institute of General Medical Sciences | R01 GM043375 | Robin Reed |
| National Cancer Institute | P50 CA165962 | Charles D Stiles |

The funders had no role in study design, data collection and interpretation, or the decision to submit the work for publication.

### Author contributions

BRP, WJG, Conceptualization, Formal analysis, Funding acquisition, Validation, Investigation, Visualization, Methodology, Writing—original draft, Writing—review and editing; LMU, Formal analysis, Validation, Investigation, Methodology; JAA, YY, PSC, EAO, DH, GW, BW, BLE, Resources, Methodology; TIZ, Data curation, Software, Formal analysis; PB, PKA, RL, Investigation, Methodology; CAN, MSB, Formal analysis, Investigation, Methodology; AT, Resources, Data curation, Software; FV, BAW, DER, GSC, Resources, Software, Methodology; SJB, Resources, Investigation, Methodology; CDS, WCH, Resources, Funding acquisition, Methodology; RR, Resources, Supervision, Funding acquisition, Methodology, Writing—original draft, Writing—review and editing; RB, Conceptualization, Resources, Formal analysis, Supervision, Funding acquisition, Investigation, Methodology, Writing—original draft, Writing—review and editing

### Author ORCIDs

Brenton R Paolella, http://orcid.org/0000-0003-3394-828X
Aviad Tsherniak, http://orcid.org/0000-0002-3797-1877
William C Hahn, http://orcid.org/0000-0003-2840-9791

### Ethics

Animal experimentation: This work was conducted in accordance with and approved by the institutional animal care and use committee (IACUC) protocols (#15-014) of the Dana-Farber Cancer Institute.

## Additional files

### Supplementary files

• Supplementary file 1. Supplementary tables. (A) Results of copy-number associated dependencies. (B) Results of CYCLOPS gene analysis. (C) Frequency of genetic *SF3B1* alterations in cancer. (D) KEGG Pathways Enriched from significant candidate CYCLOPS genes. (E) Significantly differentially expressed genes after *SF3B1* suppression in *SF3B1^neutral^* and *SF3B1^loss^* cells. (F) Summary of KEGG pathway enrichment from differentially expressed genes in each class (*SF3B1^loss^* or *SF3B1^neutral^*) from genes in Supplementary file E. (G) Significantly differentially expressed genes after enhanced SF3B1 expression. (H) PCR primer sequences used in this study.

### Major datasets

The following dataset was generated:

| Author(s) | Year | Dataset title | Dataset URL | Database, license, and accessibility information |
|---|---|---|---|---|
| Paolella B, Gibson W | 2016 | Effect of SF3B1 suppression in cancer cells with different SF3B1 copy-number levels | https://www.ncbi.nlm.nih.gov/geo/query/acc.cgi?acc=GSE81978 | Publicly available at the NCBI Gene Expression Omnibus (accession no: GSE81978) |

The following previously published dataset was used:

| Author(s) | Year | Dataset title | Dataset URL | Database, license, and accessibility information |
|---|---|---|---|---|
| Cowley GS, Weir BA, Vazquez F, Tamayo P, Scott JA, Rusin S, East-Seletsky A, Ali LD, Gerath WFJ, Pantel SE, Lizotte ph, Jiang G, Hsiao J, Tsherniak A, Dwinell E, Aoyama S, Okamoto M, Harrington W, Gelfand E, Green TM, Tomko MJ, Gopal S, Wong TC, Li H, Howell S, Stransky N, Liefeld T, Jang D, Bistline J, Meyers BH, Armstrong SA, Anderson KC, Stegmaier K, Reich M, Pellman D, Boehm JS, Mesirov JP, Golub TR, RootDE, Hahn WC | 2014 | Project Achilles | https://portals.broadinstitute.org/achilles/datasets/5/download | Publicly available at the Broad Institute Project Achilles Data Portal, version 2.4.3 (https://portals.broadinstitute.org/achilles) |

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

## Appendix

We use the following definitions:

Somatic copy-number alterations (SCNAs):

Somatic changes in the number of copies of a DNA sequence that are observed in cancer cells but not their paired germline DNA.

Gene dependency:

A gene whose suppression results in a loss of fitness for a particular cell context.

Copy-number associated gene dependency:

A gene dependency among cells that share a copy-number alteration at a given locus. In the case of cancer, most of these copy-number alterations will be somatic (SCNAs).

We also distinguish between the following classes of copy-number associated gene dependencies:

1. Cis copy-number associated gene dependencies: Here, the dependency on a gene is associated with the copy number of that gene. This can take two forms:
    a. Cancer cells can be especially vulnerable to suppression of a gene when it is amplified. This class of dependencies comprises oncogene addiction-induced dependencies, which result from amplifications of a target gene whose increased expression is important in maintaining cancer cell survival. One example is EGFR amplifications in lung cancer.
    b. Cancer cells can depend on a gene within a region of partial copy loss. For this class, reduced copy-number of the gene imparts sensitivity to further suppression of that same gene. This is the CYCLOPS (Copy-number alterations Yielding Cancer Liabilities Owing to Partial losS) class.
2. Trans copy-number associated gene dependencies: Here, the dependency on a gene is associated with the copy number of a different gene.
    a. Amplifications of a region can impart sensitivity to suppression of a gene outside the gained region. Currently, we know of no genes in this class.
    b. Copy loss of a gene can impart sensitivity to suppression of a different gene. Such dependencies can occur if deletion of one gene from a paralogous pair of genes renders cells sensitive to suppression of the remaining paralog.

## Supplemental experimental procedures

### Classification of length and amplitude for copy number alterations

For relative $log_2$ normalized copy number data analyzed from 10,570 tumors from TCGA (*Zack et al., 2013*), the following thresholds were used for copy number classification: homozygous loss $log_2$ values $< -1.2$, hemizygous loss $log_2 \leq -0.35$. For cancer cell lines used in functional studies: copy-loss cells had $log_2$ copy number $\leq -0.35$, and copy neutral cells had $log_2$ copy number $> -0.2$ and $< 0.2$.

### Analysis of the location of CYCLOPS genes on chromosome arms with known tumor suppressors

Deleted tumor suppressor genes were obtained from the combined lists from previous studies (*Beroukhim et al., 2010*; *Zack et al., 2013*). Among 8131 genes assessed by Achilles, 5546 were on chromosome arms with at least one deleted tumor suppressor gene, including 73 of the 124 candidate CYCLOPS genes. A two-sided Fisher's exact test was used to determine the significance of enrichment of CYCLOPS genes on arms shared with tumor suppressors.

## Analysis of candidate CYCLOPS gene and their overlap with *Marcotte et al. (2016)* shRNA viability screen

A complete list of genes analyzed for increased sensitivity upon partial copy loss was kindly shared by Azin Sayad and Benjamin Neel [see Table S6C in *Marcotte et al. (2016)*. A total of 8008 genes were analyzed in both studies, including 103 of the 124 candidate CYCLOPS genes we identified. We used the authors' determinations of which genes exhibited statistically significant associations between dependency and copy loss, taking $p<0.05$ as the significance threshold. The overall significance of overlap between candidate CYCLOPS genes in our study and (*Marcotte et al., 2016*) was determined using a Fisher's Exact test with a two-sided p-value.

## Analysis of gene expression across normal tissues

RNA sequencing data were downloaded from the GTEX database (http://www.gtexportal.org/home/). For every gene in the genome, we calculated the expression variance across all samples and ranked the variance against the 20 genes with the most similar average expression level. These ranks served as a nearest-neighbors normalized measure of expression variance.

## Tissue culture

Human cancer cell lines were maintained in RPMI-1640 supplemented with 10% fetal bovine serum and 1% penicillin and streptomycin and were assayed to be free of mycoplasma. Non-transformed MCF10A and HMEC cells were cultured in Mammary Epithelial Growth Medium (CC-3150, Lonza). For cells expressing tetracycline-regulated shRNAs, tetracycline-approved fetal bovine serum (Clonetech) was used. Cell lines were original purchased from ATCC or DSMZ. All cell line identities were confirmed using genetic profiling using polymorphic short tandem repeat (STR) loci at the Dana-Farber molecular diagnostics laboratory. No commonly misidentified cell lines defined by the International Cell Line Authentication Committee have been used in these studies.

## Correlation analysis of copy-loss of SF3b genes with cell dependencies upon suppression of other SF3b complex genes

We determined the relative copy number and ATARiS gene dependency scores after knockdown of each SF3b complex member across the same 179 cell lines used in the CYCLOPS analysis. We performed linear regression analysis for copy number of each SF3b complex gene with knockdown of every SF3b component. One-sided p-values were calculated for association of sensitivity to suppression with gene loss for all intra-SF3b complex comparisons. Samples were excluded if they harbored co-deletion of the two genes used to generate the correlation.

## Quantitative and reverse transcription PCR

RNA was extracted using the RNeasy extraction kit (Qiagen) and subjected to on-column DNase treatment. cDNA was synthesized with the Superscript II Reverse Transcriptase kit (Life Technologies) with no reverse transcriptase samples serving as negative controls. Gene expression was quantified by Power Sybr Green Master Mix (Applied Biosystems). Primers for all genes were determined to be equally efficient over five serial two-fold dilutions. Gene expression values were normalized to *ACTB* and the fold change calculated by the $\Delta\Delta C_t$ method. For quantification of the U2 snRNA, the above method was used except total cellular RNA was extracted with Trizol (Life Technologies). For primer sequence information see *Supplementary file 1H*.

## shRNAs targeting SF3B1

Lentiviral expression constructs for shRNA-mediated suppression of *SF3B1* were obtained through the RNAi-consortium (http://www.broadinstitute.org/rnai/public/). The clone ID's and names used in our studies are as follows: shSF3B1 #2 (TRCN0000320576), shSF3B1 #3 (TRCN0000320566), shSF3B1 #4 (TRCN0000350273), shSF3B1 #5 (TRCN0000320636). For PHF5A, the clone ID's and names used were: shPHF5A-78 (TRCN0000074878) and shPHF5A-79 (TRCN0000074879). For constitutive and inducible vectors, we respectively used the pre-existing control hairpins shLacZ and shLuciferase (*Nijhawan et al., 2012*).

## Generation of Inducible *SF3B1* shRNA expression system.

Sense and anti-sense oligonucleotides were annealed and cloned into the *AgeI* and *EcoRI* restriction sites of the pLKO-Tet-puro vector (Addgene, plasmid #21915). The oligonucleotide sequences were:

TR-shSF3B1#3 (sense) 5'-
CCGGCAACTCCTTATGGTATCGAATCTCGAGATTCGATACCATAAGGAGTTGTTTTTG,

TR-shSF3B1#3 (anti-sense) 5'-
AATTCAAAAACAACTCCTTATGGTATCGAATCTCGAGATTCGATACCATAAGGAGTTG,

TR-shSF3B1#5 (sense) 5'-
CCGGCCTCGATTCTACAGGTTATTACTCGAGTAATAACCTGTAGAATCGAGGTTTTTG,

TR-shSF3B1#5 (anti-sense) 5'-
AATTCAAAAACCTCGATTCTACAGGTTATTACTCGAGTAATAACCTGTAGAATCGAGG

## Additional information for the generation of heterozygous *SF3B1$^{loss}$* cells by CRISPR-Cas9

Oligonucleotide sequences for SF3B1$^{Loss-Cal51-1}$ were as follows: 5' CACCGCATAATAACCTGTAGAATCG (forward), 5'AAACCGATTCTACAGGTTATTATGC (reverse). pX458 was transfected with LipoD293 (SignaGen) into the diploid breast cancer cell line, Cal51.19 monoclonal cell lines were genotyped for Cas-9 induced mutations by Sanger sequencing cloned PCR products. All monoclonal lines had either no mutations or harbored biallelic mutations in *SF3B1*. The genotypes of the Cal51 CRISPR cell lines used from this method of generation were: *SF3B1$^{delT36/delT36}$* (SF3B1$^{control-Cal51-1}$) and SF3B1$^{delT36/A23fsX20}$ (SF3B1$^{Loss-Cal51-2}$). Copy number profiles from the two lines were characterized by SNP array. No SCNAs were detected as a result of single cell cloning (data not shown).

Oligonucleotides for SF3B1$^{Loss-Cal51-2}$ cells were cloned in a similar fashion as SF3B1$^{Loss-Cal51-1}$ in pX458 (with BbsI overhangs). The sequences are as follows: For the 5' guide targeting SNP, 5' CACCGCGCATTATAGATTATGGCCC (forward) and 5' AAACGGGCCATAATCTATAATGCGC (reverse). For the 3' targeting guide: 5'CACCGCGGAGTTTCATCCGTTACAC (forward), AAACGTGTAACGGATGAAACTCCGC (reverse)

## Cellular growth assays

Cells were plated in 96 well plates at 1000 cells per well. Cell number was inferred by ATP-dependent luminescence by Cell Titer Glo (Promega) and normalized to the relative luminescence on the day of plating. For short-term lentiviral infections, cells were infected 24 hr prior to plating. Relative proliferation was calculated as the fold change in ATP luciferase luminescence units from Cell Titer Glo assays relative to shLacZ expressing cells seven days after shRNA infection (*Figure 3G*).

## GFP competition assays

Oligonucleotides encoding *LacZ* or *SF3B1* shRNA#4 hairpin sequences were annealed and cloned into the pLKO.1 derivative vector TRC047 (pLKO.3pgw) and verified by Sanger sequencing. Cells were infected with serial dilutions of virus to achieve ~50% GFP-positive cells. Cells with approximately equivalent ratios of GFP-positive and negative cells were assayed by flow cytometry 3 days post infection and at subsequent time-points. The fold change in GFP+ cells was normalized to the percentage present 3 days after infection. For competition assays re-introducing exogenous *SF3B1*, we expressed a human codon-optimized *SF3B1* by lentivirus. Cells were infected as described above and treated with doxycycline two days after infection.

## Propidium iodide cell viability assays

Cells were treated with either short-term lentiviral infection or tetracycline-inducible *SF3B1* shRNAs. After treatment, cells were trypsinized and pelleted including any cells in suspension. Cells were resuspended in propidium iodide viability staining solution (1x PBS, 1% BSA, 2.5 ug/mL propidium iodide) and quantified by live-cell flow cytometry. The change in viability was normalized to the percent of viable cells quantified on the first day of the assay.

## Determination of cell cycle distribution by propidium iodide

Cells were trypsinized, washed, and fixed with ice-cold 70% ethanol for a minimum of 15 min at 4C. Cells were incubated in propidium iodide cell cycle staining solution (1x PBS, 1% BSA, 50 ug/mL propidium iodide, 100 ug/mL RNAse A) for 15 min and analyzed by flow cytometry. Debris and aggregates were gated out and cell cycle stage was quantified using Modfit (Varity Software House).

## Annexin-V apoptosis assays

Cellular apoptosis was quantified by live-cell flow cytometry using Alexa-Fluor 488 conjugated Annexin-V (Life Technologies) and propidium iodide. Cells were incubated in Annexin binding buffer containing propidium iodide (10 mM Hepes, 140 mM NaCl, 2.5 mM $CaCl_2$, 2.5 ug/mL propdium iodide) for 15 min, washed and resuspended in FACS buffer (1x PBS, 1% BSA and 50 mM EDTA). Determination of the stage of apoptosis by gating was as follows: viable cells (Annexin-V$^-$/PI$^-$), early apoptosis (Annexin-V$^+$/PI$^-$), late apoptosis (Annexin-V$^+$/PI$^+$), and dead cells (Annexin-V$^-$/PI$^+$).

## Western blotting

For denaturing protein immunoblots, cells were washed in ice cold PBS and lysed in 1x RIPA buffer (10 mM Tris-Cl Ph 8.0, 1 mM EDTA, 1% Triton X-100, 0.1% SDS and 140 mM NaCl) supplemented with 1x protease and phosphatase inhibitor cocktail (PI-290, Boston Bioproducts). Lysates were sonicated in a bioruptor (Diagenode) for 5 min (medium intensity) and cleared by centrifugation at 15000 x g for 15 min at 4C. Proteins were electrophoresed on polyacrylamide gradient gels (Life Technologies) and detected by chemiluminescence. For native western blotting, cells were washed in ice cold PBS and lysed in 1x sonication buffer (10% Glycerol, 25 mM HEPES pH 7.4, 10 mM $MgCl_2$) supplemented with protease and phosphatase inhibitors. Coomassie blue native PAGE western blots were run according the manufacturer's instructions (Life Technologies).

## Quantification of western blots by densitometry

Immunoblot exposures were quantified using ImageJ (*Schneider et al., 2012*) as previously described (http://lukemiller.org/index.php/2010/11/analyzing-gels-and-western-blots-with-image-j/). For early glycerol gradient fractions, pixel density histograms and an area under the curve (AUC) were calculated for each western blot band. Each western band AUC was normalized relative to loading control AUCs and then expressed as the fold change relative to glycerol fraction #3 (for *Figure 4F*). A minimum of 2 technical replicates per cell line (range 2–5) were performed for each of three *SF3B1^neutral^* and *SF3B1^loss^* cell lines. For *Figure 3E* and S6B-D, loading control normalized AUCs were expressed as the fold change relative to the diploid Cal51 cell line.

## SF3B1 gene expression analysis from TCGA and CCLE datasets

Relative copy number and Affymetrix expression data for *SF3B1* were downloaded from the CCLE portal (http://www.broadinstitute.org/ccle/home). TCGA breast adenocarcinoma data were downloaded from the cBioPortal (http://www.cbioportal.org/public-portal/index.do) (*Cerami et al., 2012*; *Gao et al., 2013*). For both datasets, samples lacking either gene expression or copy-number were removed. As described above, copy-loss was defined as samples with $\log_2$ normalized relative copy number of $<-0.35$, copy gain was defined as $\geq 0.3$.

## Spliceosome and DUBi drug viability assays

Cells were plated in 96 well plates at 10,000 cells per well. Relative viability was calculated as the fold change in ATP luciferase luminescence units from Cell Titer Glo assays relative to DMSO treated cells 48 hr after drug treatment (*Figure 5—figure supplement 1F*, *Figure 6H–I*, *Figure 8B,H* and *Figure 8—figure supplement 1D*).

## Immunoprecipitation

Immunoprecipitations were performed with pooled glycerol gradient fractions. The Fc region of mouse anti-SF3B1 (Medical and Biological Laboratories, D221–3) was directionally cross-linked to protein G Dynabeads (Life Technologies) using 20 mM dimethyl pimelimidate (DMP). IgG isotype controls were cross-linked and processed identically. Proteins were eluted with elution buffer (15% glycerol, 1% SDS, 50 mM tris-HCl, 150 mM NaCl pH 8.8) at 80C and subjected to western blot analysis.

## U2 snRNA 2' O-methyl oligonucleotide assays

2' O-methyl oligonucleotides complementary to the branchpoint binding region of the U2 snRNA were synthesized by Integrative DNA Technologies with the following sequences:

(Branchpoint oligo) 5'mCmAmG mAmUmAmCmUmAmCmAmCmUmUmGmA,

(control oligo) 5'mAmCmUmGmUmAmCmUmAmAmCmUmGmAmCmUmG.

Experiments were performed as described in (*Folco et al., 2012*). Briefly, mini nuclear extracts from SF3B1^control-Cal51^ and SF3B1^Loss-Cal51^ cells were generated using standard protocols (*Folco et al., 2012*). U2 snRNA-targeting and control oligos were radiolabeled with $^{32}$P-gamma ATP with T4 polynucleotide kinase. ATP was depleted from all extracts by incubation at RT for 20 min. 300 ug of total protein from nuclear extracts were added to 25 uL reaction mixtures with or without 0.5 mM ATP, 3.2 mM MgCl$_2$, and 20 mM creatine phosphate (di-Tris salt). Reactions were incubated at 30C of 5 min before adding 34 ng of either branchpoint-targeting or control oligo and subsequent incubation for another 5 min at 30C. For reactions without ATP, water was used to raise the volume to 25 uL. Samples were purified on microspin G-50 columns (GE) and loaded in a 1.2% low-melting agarose gel.

## Nuclear speckle quantification by SC-35 immunofluorescence with CellProfiler image analysis

Cells were plated on 35 mm glass bottom dishes with #1.5 cover glass (D35-14–1.5 N, In Vitro Scientific). Cells were fixed and stained with anti-SC-35 antibody (S4045, Sigma-Aldrich) at 1:1000 dilution and detected with Alexafluor488 secondary antibody at 1:1000 (Life Technologies). Nuclei were counterstained with Hoescht dye. Monochromatic images were captured under identical conditions and pseudo-colored using Photoshop. A custom image analysis pipeline was empirically adapted from a pre-existing pipeline designed for detecting H2AX foci using CellProfiler (*Kamentsky et al., 2011*). Measurements of nuclear speckles were generated from at least 15 random microscopic fields. A minimum of 100 nuclei identified by CellProfiler were used for quantitation per treatment.

## Ki67 quantification from tumor xenografts

A custom image analysis pipeline was used to systematically quantify Ki67+ cells from tumor xenografts using CellProfilier. A minimum of 3 tumors per group, totaling at least 2440 nuclei per tumor, was used to quantify the ratio of Ki67+ cells. At least five individual and random microscopic images from each tumor were analyzed.

## Luciferase splicing reporters

CMV-LUC2CP/intron/ARE (Luc-I in this manuscript, Addgene plasmid # 62858) and CMV-LUC2CP/ARE (Luc-ORF in this manuscript, Addgene plasmid # 62857) were gifts from Gideon Dreyfuss (*Younis et al., 2010*). Cells were transfected and hygromycin selected to generate stable cell lines. Cells were then treated with b-AP15 and assayed for luciferase luminescence 24 hr after treatment with the Luciferase Assay System kit (Promega). The fold change in splicing was calculated as the ratio of Luc-I to Luc-ORF after normalizing to DMSO treatment.

## RNAseq analysis of splicing by juncBase and beta binomial distribution

To identify common splicing alterations across cancer cell lines, we conducted the RNAseq splicing analyses (*Figure 6A and C* and *Figure 6—figure supplement 1D.* only in the patient derived cells lines. This was done to avoid overweighting the Cal 51 cell line from which the isogenic system was generated. We then validated samples for intron retention in both isogenic cells and those with naturally occurring copy number alterations in SF3B1 (*Figure 6B*).

JuncBASE was used to determine the spliced in/spliced out counts at each splice junction. In the following section we will use intron retention as the example for the way our beta binomial statistic works. The spliced in/spliced out counts at each junction were used to create an estimate of the percent of transcripts with the intron retained (Percent Spliced In, where the intron refers to the inclusion isoform; PSI) for each cell line. We estimated the distribution of this statistic for each cell line in each condition (with and without *SF3B1* suppression) using a beta binomial distribution in which spliced in and spliced out read counts were the $\alpha$ and $\beta$ terms, respectively. Because we only observe a finite number of reads across the junction, the PSI obtained by simple division may not be accurate. We capture the extent of this inaccuracy by calculating an estimate of the probability distribution of the true PSI, given the read counts observed. The beta binomial distribution is used in cases like this where one is attempting to obtain an estimate of the distribution of a Bernoulli random variable (true PSI) from a set of observations of trials and successes. The PSI (IR) for cell line i at intron j, is therefore estimated as:

$$IR_{i,j} = Beta\left(\alpha_{i,j}, \beta_{i,j}\right)$$

Where $\alpha = n_{reads\ intron\ j\ retained\ in\ cell\ line\ i}$ and $\beta = n_{reads\ intron\ j\ excised\ in\ cell\ line\ i}$

We then sought to quantify the probability distribution of the ratio of PSI's (PSIR) in the same cell line upon SF3B1 suppression. This ratio corresponds to the relative probability of retaining intron j in cell line i, upon SF3B1 suppression. We base this calculation on the two distributions:

$$IR_{i,j,no\ SF3B1\ suppression} = Beta\left(\alpha_{i,j,\ no\ SF3B1\ suppression}, \beta_{i,j,no\ SF3B1\ suppression}\right)$$

$$IR_{i,j,\ with\ SF3B1\ suppression} = Beta\left(\alpha_{i,j,with\ SF3B1\ suppression}, \beta_{i,j,with\ SF3B1\ suppression}\right)$$

We extend the following intuition concerning ratio distributions:

$$P(Z) = \int_0^1 \int_0^1 \frac{X}{Y}\ P(X)\ P(Y)\ dX\ dY$$

where Z=X/Y and X and Y are independent and distributed between 0 and 1.

In this case, the relative risk of including an intron for cell line i and intron j is:

$$PSIR_{i,j} = \frac{IR_{i,j,\ with\ SF3B1\ suppression}}{IR_{i,j,no\ SF3B1\ suppression}}$$

$P\left(PSIR_{i,j}\right) = \int_0^1 \int_0^1 \frac{X}{Y}\ P(x)\ P(y)\ dx\ dy$ where $x = ir_{i,j,\ with\ SF3B1\ suppression}$ and $y = ir_{i,j,\ no\ SF3B1\ suppression}$

This integral can be calculated exactly (**Pham-Gia, 2000**), by bootstrapping, or by approximation with a discrete distribution. We used the latter approach due to issues with numeric overflow in calculating the exact distribution and the computational inability to perform fine enough sampling to obtain very small p-values (ie $<10^{-9}$) using bootstrapping. To compute the discrete approximation, we iterated through a discrete set of PSI ratios spaced between 0.01 and 100. During each iteration, we calculated the joint distribution of $z_{i,j}$=x/y as above (where $z_{i,j} = PSIR_{i,j}$). We sampled x ($IR_{i,j,\ with\ SF3B1\ suppression}$) at 1000 intervals between 0 to 1 and calculated the set of y's ($IR_{i,j,\ no\ SF3B1\ suppression}$) such that y=x/z. We then calculated the joint probability of x and y for all x,y combinations and summed these values to calculate the probability of observing a given relative risk $z_{i,j}$. The discrete probability distribution of $z_{i,j}$ was then normalized to 1 to achieve the final probability distribution on the PSI ratio for intron j in cell line i upon SF3B1 suppression.

A posterior estimate of the probability distribution of the PSI ratio in SF3B1$^{loss}$ cell lines ($PSIR_{SF3B1loss}$)was then obtained by a point-wise multiplication of the distributions $PSIR_{i,j}$

$$PSIR_{SF3B1loss}(x)\ \propto\ \prod PSIR_{i,j}$$

for all cell lines with the genotype SF3B1$^{loss}$. The same procedure was used to calculate the posterior estimate of the PSI ratio for intron retention in SF3B1$^{neutral}$ cell lines $PSIR_{SF3B1neutral}$.

Finally, we calculated the p-values for the difference between the distributions $PSIR_{SF3B1loss}$ and $PSIR_{SF3B1neutral}$. This p-value represents the area of overlap of the two distributions and can be obtained by integrating proportion of the convolution of the $PSIR_{SF3B1loss}$ and $PSIR_{SF3B1neutral}$ that is greater than 0.

$$P(nh) = \int_{0}^{\infty} \left[ PSIR_{SF3B1loss}\star - PSIR_{SF3B1neutral} \right]$$

Where $P(nh)$ is the probability of the null hypothesis that the true values of $PSIR_{SF3B1loss}$ and $PSIR_{SF3B1neutral}$ are equal. Multiple hypothesis correction was performed using the Benjamini-Hochberg method (*Benjamini and Hochberg, 1995*).

