## [Decision Letter]

Thank you for submitting your article "Copy-number and gene dependency analysis reveals partial copy loss of wild-type SF3B1 as a novel cancer vulnerability" for consideration by *eLife*. Your article has been favorably evaluated by Charles Sawyers (Senior Editor) and three reviewers, one of whom, Michael R Green (Reviewer #1), is a member of our Board of Reviewing Editors. One other reviewer, Robert K Bradley (Reviewer #3), has agreed to reveal his identity.

The reviewers have discussed the reviews with one another and the Reviewing Editor has drafted this decision to help you prepare a revised submission.

Summary:

This is an interesting manuscript which identifies that cancer cells with partial copy number loss of essential genes appear to be preferentially sensitive to further depletion of these genes. These so-called CYCLOPS genes appear to be enriched in genes encoding essential proteins, including spliceosomal proteins. The authors focus on cells with copy number loss of SF3b1 and suggest that these cells are more sensitive to loss of SF3B1. This is an exciting result because it potentially expands the number of malignancies that could be treated with SF3B1 inhibitors. The experiments are generally well performed but some clarifications as described below would help improve the study as listed below. The greatest weakness of the paper is that the specificity of the deubiquitinase inhibitor for SF3B1 is not clear and the efficacy of SF3B1 binding drugs in SF3B1 partial loss cells should be studied in more detail.

Essential revisions:

1) It is unclear in Figure 5 if partial copy number of loss of SF3B1 reduces levels of U2 snRNP and/or if this occurs only following further depletion of SF3B1. It is also not clear why partial reduction of SF3b1 would reduce expression of other U2 snRNP components and U2 snRNA (unless the cells experience reduce levels of these components simply due to reduced viability upon further SF3B1 suppression).

2) The lack of efficacy of SF3B1 inhibitor drugs on cells with or without partial loss of SF3b1 is not clear from the data shown. Knockdown efficiency of the single shRNA used in Figure 5—figure supplement 1 is not shown, only 2 cell lines are studied (as opposed to the larger number of lines shown in Figure 2), and the authors have not tested the isogenic Cal51 CRISPR SF3b1 loss cell line in this experiment (only the parental Cal51 cells).

3) In Figure 6, it is not clear why intron retention is specifically focused on given that loss of SF3B1 in SFB1 copy number deleted cells should cause failure of constitutive splicing and all classes of alternative splicing. In fact, this point is suggested by the single RT-PCR example shown (MCL1 splicing; Figure 6) is a cassette exon-skipping event and not due to a change in intron retention or 3' splice site selection. Evaluating the splicing in these cells in more detail from the RNA-seq data and representative RT-PCR events would be helpful.

4) In Figure 2–Figure 3 the authors suggest that cells with partial loss of SF3B1 are more susceptible to SF3B1 downregulation however qRT-PCR data for Figure 2 are either not shown in an ideal manner to make this point and no Western blot is provided for this Figure (which may be understandable given that suppression of SF3B1 is lethal). Figure 2—figure supplement 1 gives the suggestion that SF3B1 is similarly suppressed across these cell lines following shRNA treatment but this is not the case. It would also be helpful to show the effects of complete SF3B1 downregulation on cell growth across all cell lines as these data may give the impression that knockdown of SF3B1 is tolerated in some cell types (which is not the case).

5) The selectivity of b-AP15 for SF3B1 versus other substrates is not clear. It is possible that this compound exhibits preferential effects on SF3B1 partial loss cells due to effects on SF3B1 ubiquitination in addition to ubiquitination of other substrates that are not known. Although the authors acknowledge this point, it will be important to at least examine the effect of b-AP15 on splicing/gene expression compared to knockdown of SF3B1 (as in point above) to determine whether there is a similar "signature" of altered splicing to that obtained with SF3B1 inhibition. Moreover, if it were possible to see if SF3B1 re-expression could rescue the effects of b-AP15 that would be helpful as well.

6).Figure 3 is underexposed relative to Figure 3. SF3B1 is readily visible in BT549 cells in Figure 3 but not in Figure 3.

7) The cell lines analyzed in Figure 3 should be labeled in the figure.

8) A direct comparison of SF3B1 mRNA and protein levels could be informative and help readers to understand whether reductions in mRNA levels fully explain the observed differences in protein level. The authors could illustrate this with a scatter plot or similar illustration.

9) There are several issues with the statistical analyses that should be addressed:

The authors use Fisher's exact test when testing for a difference in proportion. However, the assumptions of the test are not met, because the margins are not fixed. The authors should use the binomial proportion test instead, which does not assume that the margins are fixed. (This is a common mistake and may not change any conclusions; however, it is a statistical error that should be addressed).

The "novel statistical framework" that was used to analyze the RNA-seq data needs further explanation. Since the method is new and relevant to interpreting the data, it should be at least partially described in the main text. The description in the supplementary should also be fleshed out. For example, the authors should describe how the "risk of intron retention" relates to the more standard "psi" value (fraction of mRNAs of the parent gene containing the retained intron) in terms comprehensible to a non-statistically minded reader. An explanation of why a β-binomial distribution is superior to the more standard binomial distribution would also be helpful (e.g., from a biological perspective, the binomial distribution assumes that the probability of intron retention is fixed, while the β-binomial distribution instead assumes that the probability varies).

---

## [Author Response]

*Essential revisions:*

*1) It is unclear in Figure 5 if partial copy number of loss of SF3B1 reduces levels of U2 snRNP and/or if this occurs only following further depletion of SF3B1. It is also not clear why partial reduction of SF3b1 would reduce expression of other U2 snRNP components and U2 snRNA (unless the cells experience reduce levels of these components simply due to reduced viability upon further SF3B1 suppression).*

We primarily evaluated the effect of SF3B1 copy-loss on the U2 snRNP and SF3b complex in Figure 4 and Figure 4—figure supplement 1 without SF3B1 suppression. In contrast, Figure 5 evaluated the effect of SF3B1 copy-loss on U2 snRNP and precursor complexes after SF3B1 suppression. Quantification of large protein complexes such as the U2 snRNP (>800 kDa) can be technically challenging. However, Figure 5 provides additional data that is consistent with the conclusions of Figure 4 that SF3B1 copy-loss reduces SF3b precursor complex, but does not substantially reduce 17S U2 snRNP abundance unless SF3B1 expression is suppressed.

Given the reviewers’ concerns, we now have quantified immunoblots blots from replicate glycerol gradients from Figure 5 in the new Figure 5—figure supplement 1. There was a small but statistically insignificant trend towards decreased U2 snRNP levels in SF3B1^loss^ cells without SF3B1 suppression (Fraction 25), compared to substantial and significant loss of SF3B1 in smaller complexes such as SF3b (Fractions 4-6). We have revised the manuscript as follows:

Results:

“Consistent with Figure 4, *SF3B1^loss^* cells without *SF3B1* suppression did not have significantly lower levels of U2 snRNP in fraction 25 than *SF3B1^neutral^* cells (p=0.16, Figure 5—figure supplement 1), suggesting *SF3B1* copy-loss only reduces levels of the U2 snRNP following further depletion of *SF3B1*.”

We also observed that partial SF3B1 suppression reduced overall protein expression of U2 snRNP components SF3A3, SNRPB2 (Figure 5), and the U2 snRNA (Figure 5) in SF3B1^loss^ cells. The immunoblots and qPCR detecting these changes were performed four days after the addition of doxycycline, at a time point before the induction of apoptosis. Therefore, reduced levels of these components simply due to reduced viability upon further SF3B1 suppression is less likely. Another possibility is that the reduction in these non-SF3B1 U2 snRNP components is due to the relative instability of U2 snRNP precursors compared to their stability when assembled into the U2 snRNP. However, we have not formally evaluated this hypothesis.

We therefore now describe this caveat in the Discussion:

“We also observed decreased protein levels of SNRPB2, SF3A3 and the U2 snRNA in *SF3B1^loss^* cells after *SF3B1* suppression (Figure 5). […] Although these experiments were performed prior to detection of apoptosis, we cannot exclude the possibility that decreased U2 precursor complexes are a result of decreased cell viability beyond the level of detection.”

*2) The lack of efficacy of SF3B1 inhibitor drugs on cells with or without partial loss of SF3b1 is not clear from the data shown. Knockdown efficiency of the single shRNA used in Figure 5—figure supplement 1 is not shown, only 2 cell lines are studied (as opposed to the larger number of lines shown in Figure 2), and the authors have not tested the isogenic Cal51 CRISPR SF3b1 loss cell line in this experiment (only the parental Cal51 cells).*

We agree that additional cell lines would enhance the understanding of whether SF3B1^loss^ cells are more sensitive to chemical modulation of the spliceosome. In response, we profiled five additional cell lines with spliceosome modulators including two pairs of isogenic Cal51 CRISPR SF3B1^loss^ cell lines and HMC1-8, a SF3B1^neutral^ line with stable SF3B1 suppression. Consistent with our previous findings, we were unable to detect enhanced sensitivity of isogenic SF3B1^loss^ cells or SF3B1^neutral^ cells with partial SF3B1 suppression to spliceosome modulators and even observed a trend in the opposite direction (Figure 5—figure supplement 1).

The lack of sensitivity of cells with reduced SF3B1 protein levels generally agrees with the current mechanism of action of spliceosome modulatory drugs. These inhibitors are thought to modulate U2 snRNP function to alter splicing resulting in cell death (Corrionero et al., 2011; Folco et al., 2011; Roybal and Jurica, 2010). With similar U2 snRNP levels in *SF3B1^loss^* and *SF3B1^neutral^* cells (Figure 4), the lack of sensitivity to spliceosome modulation is not completely unexpected.

However, we identified 19 additional candidate CYCLOPS genes that are members of the spliceosome, which suggests other CYCLOPS gene dependences may be targeted by spliceosome modulators. Furthermore, copy number alterations affecting spliceosome genes may be an important consideration when evaluating the effectiveness of spliceosome modulators.

We have therefore revised the Results section as follows:

“The relative preservation of the U2 snRNP and larger complexes in *SF3B1^loss^* cells without *SF3B1* suppression suggests existing SF3b inhibitors might not exploit the specific vulnerability exhibited by *SF3B1^loss^* cells. […] We also evaluated isogenic Cal51 cells with engineered SF3B1 copy-loss and did not observe enhanced sensitivity of SF3B1^loss-Cal51^ cell lines (Figure 5—figure supplement 1).”

We also provide the knockdown efficiency of the shRNAs used in SF3B1^neutral^ cells with partial SF3B1 suppression in Figure 5—figure supplement 1 for Cal51 and Hs578T, and Figure 5—figure supplement 1 for HMC 1-8.

*3) In Figure 6, it is not clear why intron retention is specifically focused on given that loss of SF3B1 in SFB1 copy number deleted cells should cause failure of constitutive splicing and all classes of alternative splicing. In fact, this point is suggested by the single RT-PCR example shown (MCL1 splicing; Figure 6) is a cassette exon-skipping event and not due to a change in intron retention or 3' splice site selection. Evaluating the splicing in these cells in more detail from the RNA-seq data and representative RT-PCR events would be helpful.*

We agree that disruption of SF3B1 protein function is likely to impact additional classes of splicing besides intron retention or 3’ splice site selection. We have now extended our analyses to include 5’ splice site selection and cassette exon-skipping. SF3B1^loss^ cells demonstrated increased dysregulation of 5’ splice site selection with SF3B1 suppression, consistent with our prior results on intron retention and 3’ splice site selection. A larger proportion of 5’ splice sites were significantly more affected by SF3B1 suppression in SF3B1^loss^ cells (1411 junctions vs. 317, p = 9x10^-165^, Figure 6—figure supplement 1). We also observed increased dysregulation of cassette exon inclusion (Figure 6—figure supplement 1). The proportion of reads including the cassette exon at each junction changed substantially after SF3B1 suppression in SF3B1^loss^ cells, but was less substantially altered in SF3B1^neutral^ cells.

We have also validated alternative 3’ splice site junctions observed to be more frequently improperly spliced in SF3B1^loss^ cells. RT-PCR results demonstrated increased 3’ splice sites in SF3B1^loss^ cells after SF3B1 suppression, consistent with the RNA-sequencing data (Figure 6—figure supplement 1).

We have therefore revised the Results section as follows:

“The SF3b complex is known to regulate 3’ splice site selection (DeBoever et al., 2015). We therefore analyzed 30,666 alternative 3’ splice sites from the RNAseq data in *SF3B1^neutral^* and *SF3B1^loss^* cells. […] Together, these data indicate that *SF3B1* suppression more substantially dysregulates splicing of the transcriptome of *SF3B1^loss^* cells.”

*4) In Figure 2–Figure 3 the authors suggest that cells with partial loss of SF3B1 are more susceptible to SF3B1 downregulation however qRT-PCR data for Figure 2 are either not shown in an ideal manner to make this point and no Western blot is provided for this Figure (which may be understandable given that suppression of SF3B1 is lethal). Figure 2—figure supplement 1 gives the suggestion that SF3B1 is similarly suppressed across these cell lines following shRNA treatment but this is not the case. It would also be helpful to show the effects of complete SF3B1 downregulation on cell growth across all cell lines as these data may give the impression that knockdown of SF3B1 is tolerated in some cell types (which is not the case).*

We agree and in response have provided new data and revised presentation of previous data to more clearly describe the levels of SF3B1 at baseline and upon SF3B1 suppression in SF3B1^neutral^ and SF3B1^loss^ cells. We agree with the reviewers that Western blot data demonstrating SF3B1 suppression can be challenging given that complete SF3B1 suppression is lethal. Therefore, we used qPCR more frequently to provide a reliable method to evaluate SF3B1 expression at time points 2-3 days after shRNA expression before the loss of cell viability. To highlight the important differences in baseline SF3B1 expression and the extent of suppression achieved by shSF3B1 hairpins, we calculated the relative overall expression of SF3B1, normalized to the diploid cell line Cal51. Data now represent relative SF3B1 expression before and after knockdown for 11 cell lines (7 SF3B1^neutral^ and 4 SF3B1^loss^) and now present these as a main figure panel (Figure 2).

To address the level of SF3B1 knockdown at the protein level, we also performed SF3B1 immunoblots from a subset of cell lines studied. Two SF3B1^neutral^ and two SF3B1^loss^ cell lines were evaluated for SF3B1 protein expression by immunoblot two days after SF3B1 suppression using shSF3B1 #3. These Western blots were performed in parallel from cells infected and assayed by growth curves in Figure 2 that are now included in Figure 2—figure supplement 1.

We also provide new data to show the effects of complete SF3B1 downregulation on cell growth in SF3B1^neutral^ cells to reinforce the concept that complete loss of SF3B1 expression is not tolerated by any cell regardless of SF3B1 copy number status (Figure 2—figure supplement 1).

As a result, we have revised the Results section as follows:

“Upon partial *SF3B1* suppression, *SF3B1^loss^* cells exhibited significant growth defects but *SF3B1^neutral^* cells or *SF3B1^gain^*cells did not (Figure 2 and Figure 2—figure supplement 1). […] However, complete *SF3B1* suppression resulted in growth defects even in *SF3B1^neutral^* cells (Figure 2—figure supplement 1), consistent with previous work establishing *SF3B1* as an essential gene.”

We have also revised the text throughout to emphasize that it is partial SF3B1 suppression that is tolerated in SF3B1^neutral^ cells.

*5) The selectivity of b-AP15 for SF3B1 versus other substrates is not clear. It is possible that this compound exhibits preferential effects on SF3B1 partial loss cells due to effects on SF3B1 ubiquitination in addition to ubiquitination of other substrates that are not known. Although the authors acknowledge this point, it will be important to at least examine the effect of b-AP15 on splicing/gene expression compared to knockdown of SF3B1 (as in point above) to determine whether there is a similar "signature" of altered splicing to that obtained with SF3B1 inhibition. Moreover, if it were possible to see if SF3B1 re-expression could rescue the effects of b-AP15 that would be helpful as well.*

We agree that the DUBi b-AP15 is unlikely to be entirely selective for SF3B1 degradation, especially across a wide range of doses where inhibition of multiple DUBs is possible. In response, we provide new experimental evidence that suggests b-AP15 may specifically target SF3B1 to reduce SF3B1^loss^ cell viability at nanomolar concentrations that are lower than previously reported activity (D'Arcy et al., 2015). We determined the effect of b-AP15 treatment on previously identified splicing alterations that result from SF3B1 suppression to evaluate whether a similar “signature” of splicing alterations occurred during SF3B1 suppression and b-AP15 treatment. Increased intron retention was observed in SF3B1^loss^ cells, but not SF3B1^neutral^ cells treated with 550 nM b-AP15 in all three genes tested (Figure 8). We also used a luciferase splicing reporter assay to demonstrate that splicing can be inhibited in SF3B1^loss^ cells at 350 nM b-AP15 concentration (Figure 8). Of note, b-AP15 seems to stimulate splicing in the SF3B1^neutral^ cell controls in this experiment, an incidental finding that we do not understand but could be due to any number of non-specific effects of b-AP15 on these controls cells.

Importantly we also performed “rescue” experiments to demonstrate the specificity of b-AP15. Enhanced SF3B1 expression during b-AP15 treatment allowed SF3B1^loss^ cells to tolerate higher concentrations of b-AP15, similar to that of SF3B1^neutral^ cells without SF3B1 expression (Figure 8). However increased SF3B1 expression in SF3B1^neutral^ cells did not further increase cell viability during b-AP15 exposure. This suggests that non-SF3B1 effects are important considerations when exposing SF3B1^neutral^ cells to higher concentrations of b-AP15.

There is much yet to determine regarding b-AP15 and the SF3B1 CYCLOPS dependency, not least of which is whether we can more specifically target DUBs responsible for SF3B1 degradation. However, we feel that these b-AP15 DUBi studies represent a step change in our understanding of CYCLOPS dependencies in general in that they demonstrate that enhancing degradation of CYCLOPS genes through DUB inhibition can be a promising and generalizable approach to target CYCLOPS gene dependencies.

We describe these new data in the Results section:

“Treatment of cells with b-AP15 resulted in splicing alterations preferentially in *SF3B1^loss^* cells. […] Lastly, rescue experiments in which we expressed exogenous *SF3B1* increased the concentration of b-AP15 required to impact cell viability in *SF3B1^loss^* cells, but did not affect the sensitivity of *SF3B1^control^*^-Cal51-2^ cells (Figure 8).”

*6) Figure 3 is underexposed relative to Figure 3. SF3B1 is readily visible in BT549 cells in Figure 3 but not in Figure 3.*

We agree and now provide a short and long exposure of the SF3B1 immunoblot. The longer exposure in revised Figure 3 more clearly shows SF3B1 in BT549 cells.

*7) The cell lines analyzed in Figure 3 should be labeled in the figure.*

We have now included the names of the cell lines analyzed in Figure 3.

*8) A direct comparison of SF3B1 mRNA and protein levels could be informative and help readers to understand whether reductions in mRNA levels fully explain the observed differences in protein level. The authors could illustrate this with a scatter plot or similar illustration.*

We agree and provide the requested scatter plot in Figure 3. We calculated SF3B1 protein abundance relative to the diploid cell line Cal51 after normalizing for protein loading (See Appendix Methods “Quantification of western blots by densitometry”). We observed a significant correlation between SF3B1 mRNA and protein levels (p = 0.0018, R^[2]^ =0.772, regression line slope = 0.789). These data indicate that SF3B1 mRNA expression has a major influence on SF3B1 protein levels, but leave open the possibility that post-translational regulation has modest additional effects on SF3B1 protein levels.

*9) There are several issues with the statistical analyses that should be addressed:*

*The authors use Fisher's exact test when testing for a difference in proportion. However, the assumptions of the test are not met, because the margins are not fixed. The authors should use the binomial proportion test instead, which does not assume that the margins are fixed. (This is a common mistake and may not change any conclusions; however, it is a statistical error that should be addressed).*

We thank the reviewers for this helpful comment. We have now tested for differences in proportions using binomial proportions tests and have revised the p values as a result. As indicated by the reviewers, no conclusions were altered.

We have revised the Results section as follows:

“We also examined the reproducibility of the CYCLOPS analysis using a separate shRNA viability screen across 77 cell lines with a separate analytical approach (Marcotte et al., 2016) and found 49% (49/103) of the CYCLOPS genes analyzed by both datasets validated in the Marcotte dataset (p<2x10^-4^, binomial proportion test).”

“Candidate CYCLOPS genes are also enriched for essential genes. Genome-wide CRISPR viability data in human cancer cell lines identified 1,580 core-essential genes (Hart et al., 2015), including 58% (72/124) of candidate CYCLOPS genes (p<2x10^-4^, binomial proportion test).”

*The "novel statistical framework" that was used to analyze the RNA-seq data needs further explanation. Since the method is new and relevant to interpreting the data, it should be at least partially described in the main text. The description in the supplementary should also be fleshed out. For example, the authors should describe how the "risk of intron retention" relates to the more standard "psi" value (fraction of mRNAs of the parent gene containing the retained intron) in terms comprehensible to a non-statistically minded reader. An explanation of why a β-binomial distribution is superior to the more standard binomial distribution would also be helpful (e.g., from a biological perspective, the binomial distribution assumes that the probability of intron retention is fixed, while the β-binomial distribution instead assumes that the probability varies).*

We agree and have taken the following steps to enhance the clarity and interpretation of our RNAseq splicing analyses. We now provide a general description of the approach in the main Results section:

“Briefly, we calculated the ratio of percent spliced in (PSI) and spliced out read counts for each splicing junction, but accounted for the probability that any single splicing junction may not be accurately sampled in each cell line by using a β binomial distribution (see Appendix Methods).”

We also have revised our full description of RNAseq splicing analysis in the Appendix Methods. We have changed our original description of “risk of intron retention” in terms of the more standard percent spliced in “PSI” value and define the β binomial probabilities of splicing alterations in terms of PSI. We also provide an explanation of why a β binomial distribution can aid in the assessment of splicing alterations instead of a more standard binomial distribution given the relatively low numbers of reads that span splice junctions.

The text of the Appendix Methods has been modified as follows:

“JuncBASE was used to determine the spliced in/spliced out counts at each splice junction. […] We capture the extent of this inaccuracy by calculating an estimate of the probability distribution of the true PSI, given the read counts observed. The β binomial distribution is used in cases like this where one is attempting to obtain an estimate of the distribution of a Bernoulli random variable (true PSI) from a set of observations of trials and successes.”